# Grain-growth mediated hydrogen sorption kinetics and compensation effect in single Pd nanoparticles

Svetlana Alekseeva [1], Michal Strach[1], Sara Nilsson[1], Joachim Fritzsche [1], Vladimir P. Zhdanov[1,2] & Christoph Langhammer [1]✉

Grains constitute the building blocks of polycrystalline materials and their boundaries determine bulk physical properties like electrical conductivity, diffusivity and ductility. However, the structure and evolution of grains in nanostructured materials and the role of grain boundaries in reaction or phase transformation kinetics are poorly understood, despite likely importance in catalysis, batteries and hydrogen energy technology applications. Here we report an investigation of the kinetics of (de)hydriding phase transformations in individual Pd nanoparticles. We find dramatic evolution of single particle grain morphology upon cyclic exposure to hydrogen, which we identify as the reason for the observed rapidly slowing sorption kinetics, and as the origin of the observed kinetic compensation effect. These results shed light on the impact of grain growth on kinetic processes occurring inside nanoparticles, and provide mechanistic insight in the observed kinetic compensation effect.

---

[1] Department of Physics, Chalmers University of Technology, Göteborg, Sweden. [2] Boreskov Institute of Catalysis, Russian Academy of Sciences, Novosibirsk, Russia. ✉email: clangham@chalmers.se

Nanomaterials, such as nanoparticles and engineered nanostructures that find application in nanoelectronics[1], nanosensors[2], nanostructured battery electrodes[3], and as model systems in heterogeneous catalysis[4], are frequently polycrystalline. This is the consequence of that they often are created by nanofabrication methods that involve the nucleation and growth of thin films via evaporation or sputtering. Therefore, the morphology of such nanomaterials is characterized by the abundance of small crystallites or grains, which are separated by grain boundaries. In bulk systems, such structural features are well known to significantly influence physical properties[5] and make them time-dependent because grains are prone to grow via recrystallization, due to the excess energy at their boundaries, as has been reported to occur even at room temperature in macroscopic nanocrystalline Cu, Ag and Pd samples[5]. In nanomaterials and nanostructures, however, investigations of the evolution of grain morphology are rare and challenging, among other since ensemble averaging can effectively hide key features[6–14]. Therefore, the understanding of the influence of grain boundaries on kinetic processes, for example during nanoscale phase transformations, is widely lacking. This is problematic since phase transformations in nanostructured materials are a central concept in energy storage technologies like batteries[15,16] and hydrides[17], in hydrogen sensors[18,19], as well as in heterogeneous catalysis, for example in situations with metal catalyst oxidation[20–22].

To overcome this current lack of understanding, single particle experiments hold the key since they have been successfully deployed to investigate the impact of nanostructure dimensions and geometry on the thermodynamics of phase transformation processes, where they have focused on hysteresis effects[7,8,11] and the role of defects like dislocations and voids[9,10,12]. However, corresponding kinetic studies with single nanoparticle resolution are very scarce and in the few cases where the temporal evolution of the transition from one state to another was resolved[23,24], it was not quantitatively analyzed with respect to, e.g., the kinetic parameters.

In this work that fills this gap, we investigate the kinetics of the hydride formation and decomposition in individual polycrystalline Pd nanoparticles due to their importance for hydrogen storage[17], hydrogen sensors[2] and as model system for other solute-induced phase transformations in nanomaterials, and we reveal the evolution of grain morphology induced by this first-order phase transformation. This evolution we identify both as the reason for dramatically slowing kinetics after each (de) hydrogenation cycle, and as the mechanistic physical origin of the observed kinetic compensation effect in the corresponding Arrhenius parameters. Our findings thus not only clarify the effect of grain growth on the hydrogen absorption kinetics but also constitute an important contribution to the longstanding debate about the origin of such linear relations between two kinetic or thermodynamic parameters observed in chemistry and heterogenous catalysis[25,26], biology[27] and materials physics[28], where they either have been explained as statistical artefacts[29,30] or where their origin remains widely unclear[25,26,31]. Establishing a rigorous mechanistic understanding of such compensation effects, in particular for nanomaterials where corresponding studies are very rare[32,33], is therefore fundamentally important but also of high significance from an applications perspective. The latter because robust compensation effects can potentially be used for the optimization of technically important parameters of chemical transformations or reactions on nanomaterials[30,32].

## Results and discussion

We have nanofabricated arrays of 24 or 180 individual and highly polycrystalline Pd nanoparticles by electron-beam lithography

onto an oxidized silicon wafer for hydrogenation experiments and onto electron-transparent silicon nitride membranes for transmission electron microscopy (TEM) imaging (Fig. 1a). The resulting nanodisks all have a diameter of 200 nm and a height of 30 nm with an average crystallite size of ~10 ± 2 nm (Supplementary Fig. 1). For all experiments, they are used starting in the freshly grown state, without any prior annealing. To resolve the fast hydrogen sorption kinetics for all single particles simultaneously, we have employed multiplexed single-particle plasmonic nanoimaging microscopy[34], in which optical contrast is generated by measuring changes in spectrally integrated scattering intensity.

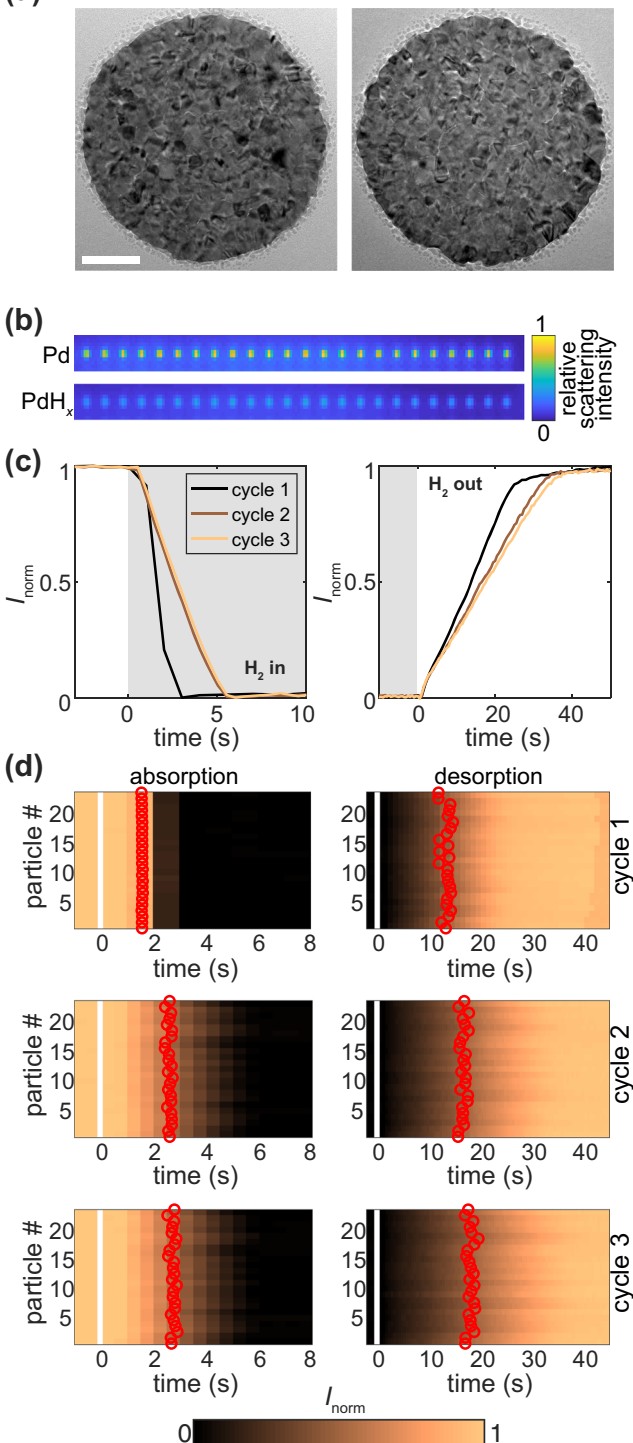

**(a)**

**(b)**

**(c)**

**(d)**

**Fig. 1 TEM of Pd particles, experimental procedure and single particle kinetics. a** Transmission electron microscopy images of as-deposited Pd particles, which reveal the polycrystalline structure with an average grain size of ~10 nm (see also Supplementary Fig. 1). Scale bar is 50 nm. **b** CCD image of the 24 Pd nanodisks in the metallic (Pd) and hydride (PdH$_x$) state. The particles are revealed as diffraction-limited spots in the scattering image, which upon hydrogen absorption loose intensity. The color scale is linear and the same for both images. **c** Dark-field scattering intensity signal over time normalized to the maximum value for one of the 24 Pd nanoparticles for hydrogen absorption (left) and desorption (right) for the 1$^{st}$, 2$^{d}$ and 3$^{d}$ (de)hydrogenation cycles (color). Gray and white boxes depict the on- and off-set of a 150 mbar H$_2$ pulse, respectively, at time $t = 0$. **d** Scattering intensity over time normalized to the maximum value (color bar) as in **c** but shown for all 24 particles simultaneously, where each horizontal line represents a measurement from a single particle. Measurements are shown for absorption (left) and desorption (right) for the 1$^{st}$, 2$^{d}$ and 3$^{d}$ cycle. White vertical lines at $t = 0$ indicate introduction or removal of H$_2$ to/from the system, respectively. Red circles indicate individual $t_{50}$ values of the particles. For extraction procedure of $t_{50}$ see Supplementary Fig. 3.

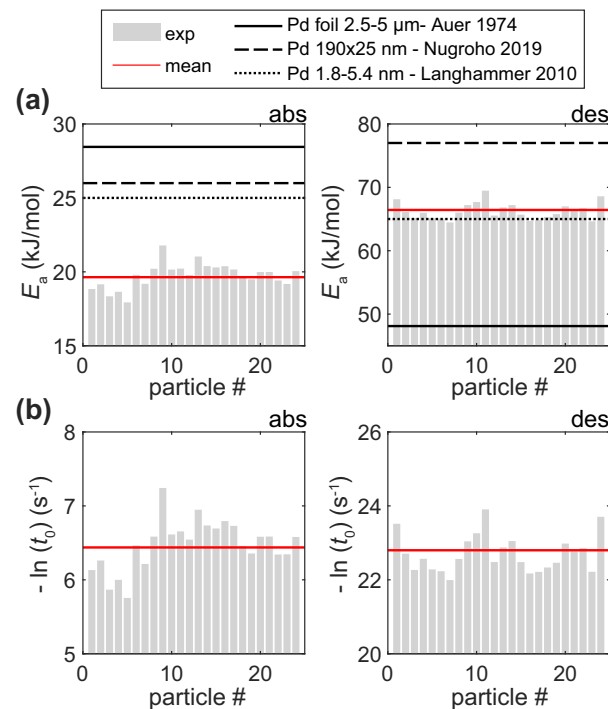

**Fig. 2 Arrhenius parameters of individual Pd particles. a** Apparent activation energy and **b** pre-exponential factor for absorption (left) and desorption (right) extracted for the 24 individual Pd particles. The light gray bars are data from LLS fits, with the mean value indicated as red solid lines. The data were extracted from a first set of measurements, starting with the pristine as-grown sample, as shown in Supplementary Fig. 4, and included a full set of consecutive measurements across the $T$-range of 338 to 303 K. Black lines in the upper panel denote $E_a$ values for annealed Pd nanoparticles[39] of 1.8–5.4 nm in size (dotted), annealed Pd disks[38] of 190 nm in diameter and 25 nm in height (dashed) and Pd foil[37] of 2.5–5 μm in thickness (solid). Analogous figure with NLLS data can be found in Supplementary Fig. 9. Goodness-of-fit statistics for both methods are given in Supplementary Fig. 10 and the corresponding Arrhenius plots and nonlinear fits to the raw data in Supplementary Figs. 5 and 6.

Accordingly, we rely on the linear optical contrast between neat Pd, Pd-H in the solid solution α-phase, in the α+β coexistence region, and in the pure β-phase (hydride)[35,36]. Corresponding charge-coupled device (CCD) camera images of a 24 Pd nanoparticle array in the metallic and hydride state confirm the anticipated sizeable reduction in scattering intensity upon hydride formation (Fig. 1b). The determination of rate constants in kinetics measurements can depend on the speed at which the pressure changes in the sample chamber. In our experimental setup (Supplementary Fig. 2), we use series of valves at the inlet and outlet sides of the measurement chamber, which allow rapid change of pressure during both absorption and desorption steps, with a time constant of <0.5 s. In this way the impact of the speed of pressure change on the measured kinetics is nearly negligible. Measuring the (de)hydrogenation of the array three times in sequence, by applying 150 mbar H$_2$ pulses at 303 K, we observe that the kinetics for both absorption and desorption slow down considerably with each cycle (Fig. 1c, d). Moreover, the time constant for the very first hydrogen absorption is essentially identical for all 24 particles but starts to scatter more between them already for the first desorption step, hinting at a hydrogen-induced evolution of increasingly more single-particle specific response upon repeated cycling (Fig. 1d). At the same time, we also note that the time constant of our instrument (< 0.5 s) and the temporal response of the particles during the first hydrogenation cycle ($t_{50} \approx 1.5$ s) are close enough that we cannot completely exclude some convolution of these two factors and this being a contributing factor for the observed very similar response of the individual particles during the first – and fastest – hydrogen absorption cycle.

To identify the reason for this behavior, we measured the temperature dependence of the characteristic time to reach 50% signal intensity change, $t_{50}$, upon H$_2$ ab-/desorption in the temperature range from 338 to 303 K in 5 K steps simultaneously for all 24 particles (Supplementary Fig. 4). From this data set, we then extracted the corresponding apparent activation energies for each particle, by assuming that $t_{50}$ can phenomenologically be represented in the Arrhenius form by

$$t_{50} = t_0 \exp(E_a/RT) \qquad (1)$$

where $t_0$ and $E_a$ are the apparent pre-exponential factor and activation energy, respectively, $R$ is the gas constant, and $T$ is the temperature. By constructing the conventional Arrhenius plot

ln($t_{50}$) versus $1/T$ and applying linear least squares (LLS) regression (see Supplementary Fig. 5 for nonlinear least squares (NLLS) regression and Supplementary Figs. 6–8 for LLS regression), we can extract $E_a$ and $t_0$ from the slope and the intersection with the $y$-axis, respectively. The correspondingly found apparent $E_a$ are similar for all 24 probed individual particles and in reasonable agreement with the literature for a Pd foil[37], ensembles of Pd nanodisks[38] and 2–5 nm Pd nanoparticles[39] (Fig. 2). However, comparing the trends for $t_0$ and $E_a$ between individual particles, we observe *compensation* between the two parameters, which we will discuss in detail below.

To first investigate the mechanistic origin of the observed slowing kinetics upon hydrogen cycling, we performed a series of intermittent grazing incidence X-ray diffraction (GIXRD) experiments after 0, 1, 2, 3, 6, 11, 33 and 64 (de)hydrogenation cycles on a sample covered with a large-area array of Pd nanodisks (Fig. 3a). Deriving the average crystallite size after each step reveals (see Methods section for details on size extraction procedure) a power-law (with low exponent, $1/m \ll 1$) growth with increasing number of (de)hydrogenation cycles (Fig. 3b),

$$\langle R_{cr} \rangle = (\langle R_{cr} \rangle_0^m + A^* t)^{1/m} \qquad (2)$$

where $\langle R_{cr} \rangle_0$ is the initial average crystallite size, and $A^*$ and $m$ are parameters. This law has been widely used in studies focused

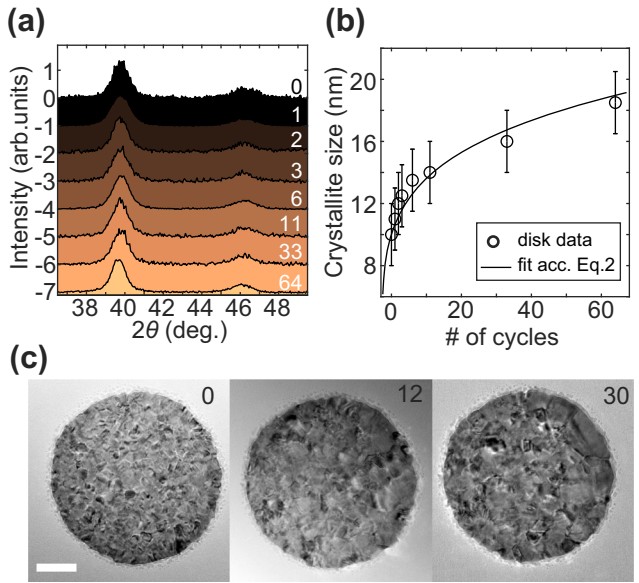

**Fig. 3 Grain growth characterization by GIXRD. a** GIXRD patterns of Pd nanoparticles (de)hydrogenated *N* times, with cycle number *N* indicated in the legend to the right and cycle temperature history as depicted in Supplementary Fig. 4. Patterns are shifted vertically from each other for clarity. **b** Average crystallite size extracted from GIXRD measurements as function of *N*. Error bars were estimated based on statistics from multiple measurements for each sample at different incidence angles. The black curve is the fit to the data according to Eq. 2 with *m* = 5. **c** TEM image series of the same representative single Pd particle after 0, 12 and 30 (de) hydrogenation cycles, corroborating substantial grain growth. Scale bar is 50 nm.

on grain growth in macroscopic samples[5]. For such samples, the simplest theory[40] focused exclusively on the interface curvature as a driving force for grain growth predicts *m* = 2, whereas experiments and more advanced models[5] usually show *m* in the range from 3 to 5 (sometimes up to 12, which then means that the growth is nearly terminated). This wide range is the consequence of that the exact kinetics of grain growth depend on a multitude of factors in the material (e.g., triple junctions, non-equilibrium vacancies and nanovoids, and complex diffusion of atoms)[5], and also reflects the fact that a detailed full-scale interpretation of this process and the exact mechanistic meaning of experimentally derived values of $A^*$ and *m* is still lacking. Fitting Eq. (2) to our GIXRD data yields good agreement for *m* = 5 (Fig. 3b), a value typical also in macroscopic nanocrystalline samples[5]. For our case at hand, this means that we indeed are observing a grain-growth process with kinetic characteristics common for nanocrystalline materials. To further corroborate the GIXRD results, we performed TEM characterization of the same single nanoparticle before cycling, and after 12 and 30 (de) hydrogenation cycles, respectively, which confirms the significant hydrogen sorption-induced grain growth (Fig. 3c).

To evaluate the impact of the identified (de)hydrogenation-induced grain growth on the sorption kinetics in more detail, we extracted the $t_{50}$ values for hydrogen absorption and desorption for all 24 single nanoparticles after 0, 2, 3, 11, 19, 33 and 34 (de) hydrogenation cycles, and plotted the corresponding histograms in Fig. 4a and b, respectively. Strikingly, for absorption, the average $t_{50}$ increases from 1.6 to 10.9 s, and for desorption from 13.4 to 83.7 s. Furthermore, we notice a significant broadening in the $t_{50}$ distribution with increasing number of cycles, which indicates a dramatically increased particle individuality imposed by the single particle specific evolution of the grain structure. The

key factor here is that with increasing number of cycles the average number of grains becomes much smaller. The measure of the effect of this factor on $t_{50}$ is the ratio of the width of the distribution of this number to the average number. According to the Poisson distribution, this ratio increases with decreasing the average number of grains. From this perspective, the system becomes more heterogeneous with increasing number of cycles, and it results in a significant broadening in the $t_{50}$ distribution. In addition, the observed structural evolution not only is reflected in the single particle sorption kinetics, but also in the corresponding thermodynamics, where single particle pressure-composition isotherms reveal significant increase in hysteresis upon cycling (Supplementary Fig. 11a, b), thereby corroborating recent results correlating grain boundary length with hysteresis width (Supplementary Fig. 11c)[11]. Finally, from an application perspective, the found evolution of structure and kinetics over time identifies a potentially critical feature responsible for the temporal evolution of the structure of (Pd-based) nanoparticles used in hydrogen storage systems, and a so far unexplored design rule for the development of ultrafast hydrogen sensors in par with the US DoE response time target[18,41], namely the optimization and stabilization of grain structure.

To further shed light on the underlying mechanism responsible for the dramatic impact of grain structure on the absorption kinetics of the individual nanoparticles, we recall that the absorption of $H_2$ occurs via dissociation on the Pd surface, penetration of H into the subsurface region, and further diffusion into the particle. Mechanistically, the growth of a hydride inwards in a nanoparticle can thus to a first approximation be described in a core-shell fashion, and it has been shown that the shape of the absorption kinetics depends on the evolution of lattice strain induced by hydride formation[42], whereas the timescale of the process can be represented as[39]

$$t_{50} = A \, D^z \qquad (3)$$

where *A* is a constant that depends on the diffusion coefficient of H in the host, *D* is the particle diameter, and *z* is the corresponding exponent. For conventional diffusion-limited growth and a perfectly homogeneous shell *z* = 2. However, in reality, *z* is larger and for single-crystalline Pd nanoparticles in the sub-10 nm size range *z* = 2.9 has been found[39]. In the polycrystalline Pd particles under consideration here, the highly abundant grain boundaries provide an additional pathway for hydrogen diffusion. To understand its relative role compared to H-diffusion through the grains, it is instructive to compare the activation energies of hydrogen diffusion through a homogeneous Pd hydride shell layer and along a grain boundary. For the former channel, experiments yield $E_d$ = 0.24 eV[43]. For the latter channel, $E_d$ is unknown but, realistically assuming that (111) facets constitute a sizable fraction of the grain boundary structure, it can be estimated as $E_d$ = 0.09 eV (experiment[44]) or 0.12 eV (DFT calculations[45]). The difference between these energies, ≃ 0.14 eV, is significant and indicates that grain boundary diffusion is fast[46,47], and indeed may account for the observed dramatically increasing $t_{50}$ values upon grain growth. Accordingly, it is reasonable that rapid H diffusion along grain boundaries in the early stages of hydrogenation leads to the formation of a thin hydride layer at the boundaries that surround individual grains, with hydride formation then proceeding in the same way as in a small single crystal, within each grain (Fig. 4e, f). In this model, the absorption kinetics can be described by identifying *D* in Eq. (3) with the average grain size of the polycrystal. The corresponding fit to our experimental single nanoparticle data yields *z* = 4.5 (Fig. 4c). This exponent is somewhat larger than the *z* = 2.9 obtained for small single crystalline Pd particles[39]. However, the discrepancy can likely be explained by a different/higher level of tensile strain induced by the thin hydride layer within each grain in

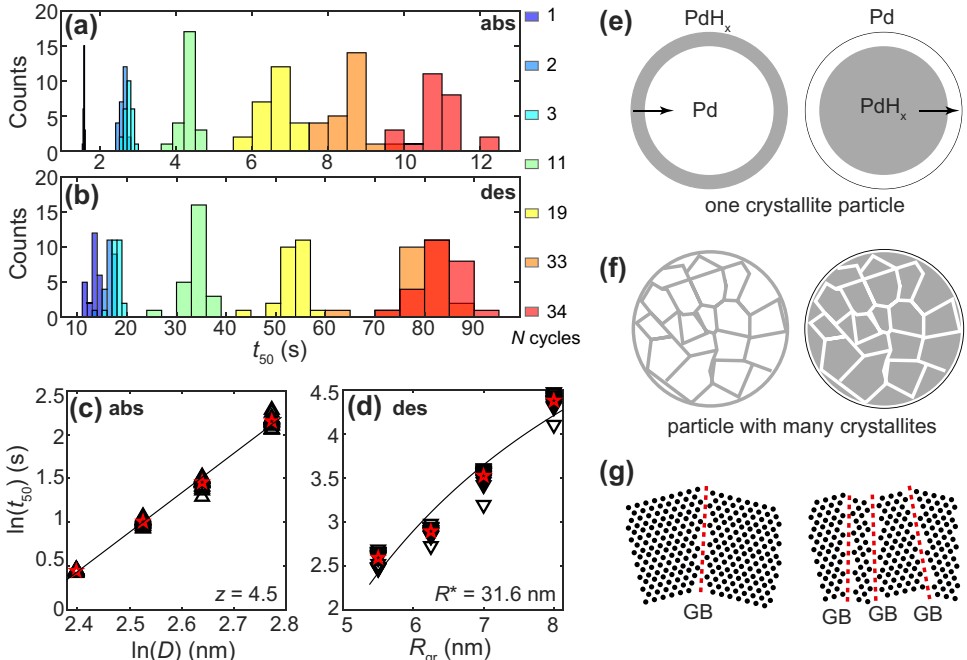

**Fig. 4 Grain growth influence on $t_{50}$ and schematics of (de)hydrogenation process. a** Distribution of $t_{50}$ values corresponding to absorption and **b** desorption for $N$ (color code) (de)hydrogenation cycles measured at 303 K. Note the dramatic deceleration of the sorption kinetics and widening of the $t_{50}$ distribution upon cycling. **c** Power-law scaling of $t_{50}$ and crystallite diameter ($D$) of 24 individual Pd particles. The linear fit corresponds to $z = 4.5$ according to Eq. 3. **d** $\ln(t_{50})$ as a function of grain radius ($R_{gr}$) for dehydriding. The solid line represents the best fit according to Eq. 4, resulting in the corresponding parameter $R^* = 31.6$ nm. Red stars in **c** and **d** represent the average value for 24 individual particle data (upward and downward pointing triangles, respectively). Slowing down of the kinetics is observed for all measured temperatures for both absorption and desorption (Supplementary Fig. 12). **e** Schematics of hydride formation (left) and decomposition (right) for a particle consisting of a single crystallite and **f** a particle containing many crystallites. The arrows show the direction of the (de)hydrogenation process (inward or outward). Gray and white areas depict the hydride and metal state, respectively. **g** Schematics of the atomic structure at the surface of a particle for the case with few (left) and many (right) grain boundaries (GB) indicated by red dashed lines, where the presence of many GBs leads to an atomically rougher surface with more low-coordinated atoms.

the case of the polycrystal, compared to a small single crystal particles[11,48].

Next, we move on to the discussion of the impact of grain-growth on the hydride decomposition and $H_2$ desorption kinetics. To this end, it is commonly accepted that Pd hydride decomposition kinetics are controlled by associative desorption from the metal-gas interface[49]. In other words, the diffusion-mediated supply of H to this interface from the inside of the Pd particle is not rate limiting. For the case of our polycrystalline particles, this means that an additional grain-boundary diffusion channel is not expected to influence the kinetics. In fact, $H_2$ desorption takes place from the surface of the grains at the gas-nanoparticle interface. In analogy to small single crystalline particles[39], the desorption rate associated with a single grain at the surface of the polycrystalline particle can therefore be represented as $w = r\,s\,\exp(R^*/R_{gr})$, where $s$ is the area of the grain gas-metal interface, $r$ is the desorption rate per unit area in the limit when surface strain effects are negligible, and $\exp(R^*/R_{gr})$ is the factor taking surface strain into account ($R_{gr}$ is the grain radius, and $R^*$ is a constant proportional to the surface strain). The desorption rate associated with a polycrystalline nanoparticle is accordingly given by $W = r\,S\,\exp(R^*/R_{gr})$, where $S$ is the area of the gas-metal interface for the entire particle. If we now for a first approximation initially assume that the atomic surface structure does not depend on grain size, $S$ becomes grain size independent, and the timescale of hydrogen desorption from a polycrystalline Pd nanoparticle can be represented as

$$t_{50} = B\exp(-R^*/R_{gr}) \qquad (4)$$

where $B$ is a constant proportional to the $V/S$ ratio ($V$ is the

nanoparticle volume). The corresponding fit to the experimental single particle data yields $R^* = 31.6$ nm (Fig. 4c). This value is significantly larger than $R^* = 3.9$ obtained earlier for small single crystalline Pd nanoparticles[39]. This significant difference between the two $R^*$ values may either indicate a stronger dependence of the desorption rate on $R_{gr}$ in a polycrystalline particle, compared to a single crystalline one, or it may indicate that the atomic structure of the polycrystalline particle surface depends strongly on $R_{gr}$, and that our initial assumption of the grain size independence of $S$ therefore is not correct. Physically, the second scenario is very likely because the extent of atomic-scale roughness is expected to decrease with increasing $R_{gr}$, since fewer grain boundaries characterized by an open atomic structure and by Pd atoms with low coordination will be present at the surface of the particle, where $H_2$ desorption occurs (Fig. 4g).

Following this line, the faster desorption kinetics we observe in the small grain regime can be understood from an energetics point of view, where it is known that the activation energy for $H_2$ desorption from more open Pd(211) and Pd(100) faces is lower than from Pd(111), with a computed energy difference of about 0.1 eV[50,51]. This concept can be further supported by XRD studies of thin Pd films, which show a weakening/disappearance of the (200) reflection and intensity increase of the (111) reflection upon repeated $H_2$-cycling[52,53]. This indicates an energetically more favorable packing of Pd atoms and consequent reorientation in the out-of-plane (111) direction upon stress-strain cycling induced by (de)hydrogenation, which is also reasonable to expect for our polycrystalline particles.

As a final aspect, we also notice that other types of structural defects, such as impurities, vacancies and dislocations have been

reported to affect hydrogen sorption kinetics in Pd systems like thin films[54,55]. As the main difference we find, however, that the initial grain size in our case is significantly smaller, which has a number of key consequences. The first one is that dislocations not associated with grain boundaries, i.e., within the crystallites, are very unlikely to play a role since they energetically are not allowed in crystallites in the (sub-) 10 nm range[8,56,57]. Similarly, the number of vacancies is expected to be low compared to grain-boundary-related defects and impurities are very unlikely to play a critical role due to the high purity source material used to grow the Pd nanoparticles.

Having established that grain growth in the Pd nanoparticles is induced by hydrogen sorption and that this process significantly impacts the corresponding kinetics, it is now interesting to investigate how it affects the Arrhenius parameters and compensation effects, as identified in Fig. 2. To do that, we simultaneously measured $t_{50}$ for a large number of freshly prepared single Pd nanoparticles using a set of multiple temperature sweeps to derive the apparent $E_a$ and $t_0$ for each $T$ sweep. For one sample comprising 180 particles, we consistently sweep $T$ from highest to lowest ($T_{down}$), for another sample (180 particles) from lowest to highest ($T_{up}$), and for a third sample (24 particles) we use a mix of $T$ sweep directions ($T_{mix}$ – down, up, up, down and up).

Focussing on hydrogen desorption (Supplementary Fig. 13 for identical analysis of absorption data), in Fig. 5a-c we present modified Cremer-Constable plots[31], which characterize the compensation effect by depicting $E_a$ for each individual particle derived from the Arrhenius analysis (Supplementary Fig. 6–8) of the different sets of $T$ sweeps plotted versus $\ln(t_0)$ multiplied by the ideal gas constant. The slope of the plots corresponds to the so-called isokinetic temperature, $T_{isokin}$, at which all the particles in the specific measurement set have the same rate of reaction[30]. Furthermore, when $T_{isokin}$ is close to the harmonic mean temperature of the experiment ($T_{hm} = 320$ K in our case) it is argued in the literature that any observed compensation effects can be attributed to statistical and/or experimental errors[29,30]. Interestingly for our data, however, for all three $T$-sweep directions we observe a $T_{isokin}$ that falls outside the $T$ range of the experiment, despite the narrow experimental $T$ range. To demonstrate the extent of correlation between the derived Arrhenius parameters, in the insets of Fig. 5 a–c for each of the $T$ sweep directions, we also show the normalized average $E_a$ (left y-axis) together with normalized $t_0$ at the average experimental temperature $<T> = 320$ K (right y-axis). We then plot the average $E_a$ for each set of 180 and 24 single particles, respectively, and for all $T$-sweeps as function of number of $T$ sweeps (Fig. 5d). As the key result, we find a consistent decrease of $E_a$ when $T$ is decreased within a sweep, and consistent increase of $E_a$ when $T$ is increased.

These results can be rationalized by taking the found grain-growth induced kinetics slowing effect into account. To generate a better understanding of how an activation energy derived by Arrhenius analysis may be affected by a grain growth induced slowdown of the apparent sorption kinetics and how this effect depends on the direction of the $T$-sweep used, we present the following thought experiment. Let's take the average $E_a$ and $t_0$ values from the first set of measurements shown in Fig. 2 in the main text (66 kJ/mol for $E_a$ and –22.5 for $t_0$ at desorption) and assume that these parameters represent an ideal situation without grain growth. Based on these values, we can generate a set of $t_{50}$ values that perfectly fit these Arrhenius parameters within the temperature range of our experiments (Eq. 1, with $R_{square} = 1$). In Fig. 5d they are presented as black circles. Let us now include grain-growth in our scenario. During the $T$-sweeps, irrespective of their direction, after each cycle desorption becomes slightly slower due to grain growth (cf. Fig. 4b). However, the extent of

slowing is manifested differently depending on the direction of the $T$-sweep. For $T$-sweeps down, we see significant slowing of $t_{50}$ due to the fact that we move towards lower temperatures, where intrinsically slower kinetics at lower $T$ coincide with additional slowing due to grain growth upon cycling (blue triangles in Fig. 5e). In contrast, for $T$-sweeps up, the slowing is less pronounced since in the beginning of the experiment everything is faster, and for measurements at higher temperatures it is intrinsically faster as well (red triangles in Fig. 5e). Accordingly, considering the way an Arrhenius analysis is constructed (Fig. 5f), for the $T$-sweep with increasing temperature the extracted apparent $E_a$ will be somewhat smaller compared to a situation without grain growth (Fig. 5g). Following this line, when we use $T$-sweeps with decreasing temperature the situation is opposite, and the derived $E_a$ will be somewhat larger compared to a scenario without grain growth. Hence, the impact of grain growth induced slowing of the sorption kinetics after each cycle can also provide an explanation for the observed oscillations in the Arrhenius parameters (Fig. 5d) when the $T$-sweep directions are alternated between up and down directions. In particular, one can see that $E_a$ are higher for the 1st and 4th $T$-sweeps, where the sequential measurements within each sweep were performed by decreasing temperature and lower for the 2nd and 3rd and 5th $T$-sweeps where the measurements were performed by increasing temperature. Finally, since the amount of grain-growth induced by a single hydrogenation cycle decreases with an increasing total number of cycles (cf. Fig. 3b), we see the convergence of $E_a$ towards a value that is independent from the direction of the $T$-sweep (Fig. 5d).

The above observations have a number of key consequences. First, the measured Arrhenius parameters are "apparent" in the sense that they characterize in a lumped way not only the process of hydrogen sorption, but also the effect of hydrogen-sorption-induced grain growth. Secondly, this grain growth provides a mechanistic explanation for the observed compensation in the Arrhenius parameters. To further corroborate this conclusion, we employ a framework recently introduced by Griessen et al.[30], which establishes the likeliness of a non-statistical origin of compensation effects by calculating a compensation quality factor, CQF. In our case, it indeed reveals CQF values of 0.57 and 0.83 for desorption at $T$-sweeps up and down, respectively and thereby confirms the non-statistical origin of the observed compensation (for details of the analysis see SI Section 12 and Supplementary Figs. 14–17, and for general discussion of the compensation effect in physicochemical processes, see ref. [26]).

Finally, it is interesting to see if the large number of 180 single particles measured simultaneously for the $T_{up}$ and $T_{down}$ sweep samples yields additional information. Therefore, for each individual particle, we plot a slowing factor (SF) versus the individual particle's apparent $E_a$ obtained from the 1st $T$-sweep (Fig. 6a, b). Here, SF is defined as the ratio of the last $t_{50}$ measured at 303 K to the first $t_{50}$ measured at 303 K. This analysis indeed reveals a correlation between single-particle-specific SF and $E_a$, where $E_a$ increases as particles sorb $H_2$ more slowly. Mechanistically, this is the consequence of the specific evolution of the grain structure within each single nanoparticle, where grain growth occurs at individual rates that are dictated by the specific grain structure in each particle directly after Pd evaporation, and where the effect is strongest for the first few hydrogenation cycles (cf. Fig. 3b). Therefore, upon further cycling, grain growth per cycle is reduced and becomes more uniform for all particles, and accordingly the correlation between SF and $E_a$ becomes less pronounced (Supplementary Fig. 18). In view of these findings, it is also relevant to revisit the compensation effect observed in Fig. 2, which we now clearly can assign to be the consequence of single particle-specific grain growth. However, in the corresponding data set of 24

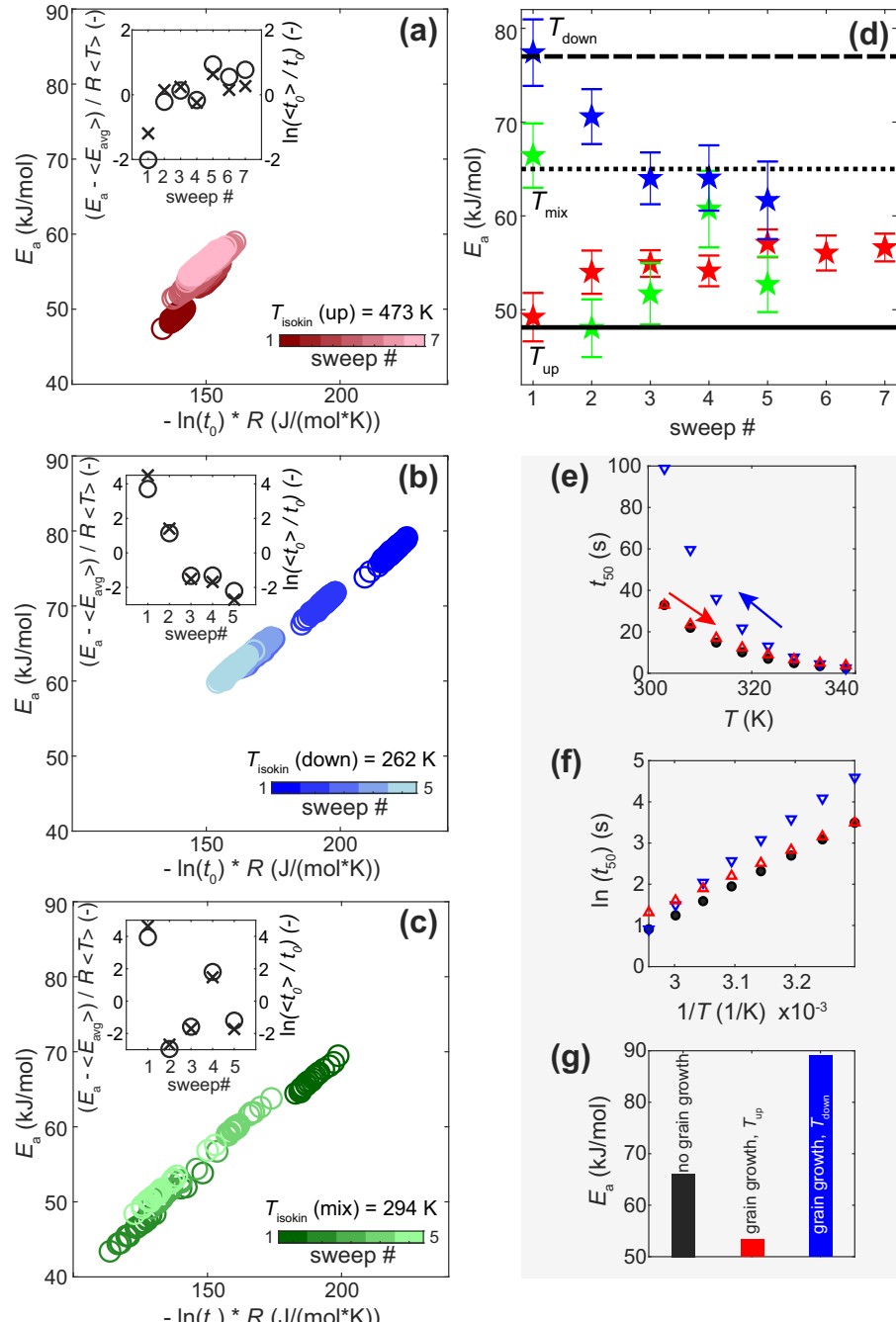

**Fig. 5 Compensation effect in single particle Arrhenius parameters. a** Modified Cremer–Constable plots at desorption for three samples measured with different $T$-sweeps: up—red; (**b**) down— blue and **c** mixed—green. Each subsequent $T$-sweep in a set is colored as a lighter shade of main color (colorbar). As a measure of compensation, insets show the normalized change in activation energy, $(E_a-\langle E_{avg}\rangle) / RT$ (where $E_a$ is the average value for 180 ($T_{up}$ and $T_{down}$) or 24 particles ($T_{mix}$) in each sweep, $\langle E_{avg}\rangle$ is the average of the entire set, black circles), and logarithm of the pre-exponential factor ($\ln(\langle t_0\rangle/t_0)$, where $t_0$ is the average value for the particles in each sweep and $\langle t_0\rangle$ is average of the entire set, black crosses) as function of measured $T$-sweeps. $T_{isokin}$ for all data measured with each of the $T$-sweep directions ($T_{mix}$, $T_{down}$ and $T_{up}$) lumped together was calculated according to ref. [30]. **d** Average $E_a$ of the particles for each of the measured sweep directions (colored stars: $T_{up}$—red, $T_{down}$—blue and $T_{mix}$—green). Black lines denote $E_a$ values at desorption for annealed Pd nanoparticles[39] of 2–5 nm in size (dotted), annealed Pd disks[38] of 190 nm in diameter and 25 nm in height (dashed) and Pd foil[37] of 2.5–5 μm in thickness (solid). Error bars indicate 95% confidence bounds of the Arrhenius fit. See analogous figure for absorption in Supplementary Fig. 13. In **e**–**g** we simulate three scenarios for the derivation of the apparent activation energy at desorption: the ideal case without grain growth (black circles), grain growth with $T$-sweep up (red triangles) and grain growth with $T$-sweep down (blue triangles): (**d**) $t_{50}$ vs. $T$, (**e**) Arrhenius plots of $\ln(t_{50})$ vs. $1/T$ and **f** corresponding apparent activation energy for each scenario.

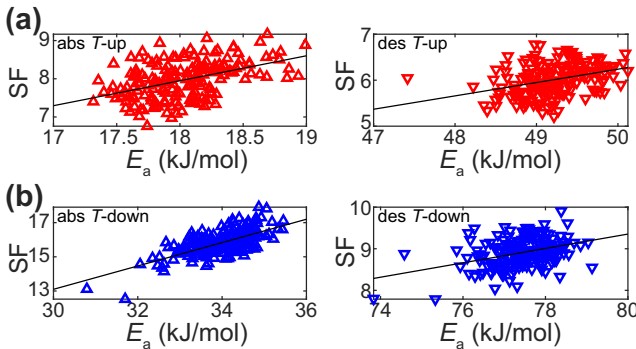

**Fig. 6 Correlation of single particle kinetics slowing factor with activation energy. a** Slowing factor, SF = $t_{50}$ (last) / $t_{50}$ (first) at 303 K versus activation energy, $E_a$, obtained from the 1st $T$-sweep (see Supplementary Fig. 18 for all other $T$-sweeps) for samples measured according to scheme $T_{up}$ and **b** $T_{down}$ at absorption (left) and desorption (right). For $T_{up}$ SF = $t_{50}$ (53)/ $t_{50}$ (1) and $T_{down}$ SF = $t_{50}$ (43)/ $t_{50}$ (1), where numbers in parentheses indicate the cycle number at which the corresponding last and first measurement at 303 K was performed for each sample. Black lines indicate linear regression to the data.

particles only, the correlation was not conclusively revealed and leads to a low CQF value due to insufficient statistics (Supplementary Fig. 19), again highlighting the importance of appropriate experiment and sample design, when studying compensation effects in nanomaterials.

In conclusion, using multiplexed plasmonic nanoimaging microscopy of large arrays of individual Pd nanoparticles combined with GIXRD and TEM characterization, we have demonstrated that substantial grain growth is induced by each (de) hydrogenation cycle. This growth significantly slows down the kinetics of hydrogen sorption, and makes them more and more single-particle specific due to individualistic grain morphology evolution in each particle. Mechanistically, we identify diffusion of hydrogen along, and tensile strain induced by a hydride layer at, the grain boundaries as the main factors influencing the absorption kinetics, while for desorption the energetics and abundance of grain-boundary sites located at the nanoparticle surface are critically determined by the grain growth. As the second key result, we have unambiguously identified the grain-growth process as the mechanistic origin of the observed kinetic compensation effect. In other words, we have observed convolution between the conventional Arrhenius effect of temperature and the effect of grain growth on the kinetics. This constitutes an important contribution to the longstanding debate about the origin of such compensation effects, since it sets our work apart from previous reports, which either explain them as statistical artefacts[29,30] or don't provide clear mechanistic explanations despite indications of the observed effects having a physical origin[25,26,31]. Since this breakthrough was enabled by highly parallelized single particle measurements, our work also highlights the critical importance of careful experiment design and statistically relevant and sufficiently large single particle data sets to reveal such effects in nanomaterials, since they otherwise will be averaged out in the ensemble or be lost between different experiments.

Looking forward, our findings have direct implications for the fundamental understanding of metal-solute interactions and unravel a so far unexplored handle for the rational design of nanomaterials that find application in solid-state hydrogen storage and hydrogen sensors with ultrafast response and long-term stability. Specifically, the presence of grains can be useful here, and accordingly it becomes clear that single crystalline structures

obtained, for example, by colloidal synthesis very likely constitute inferior solutions with respect to hydrogen sorption kinetics. Therefore corresponding development should be steered towards highly polycrystalline structures, ideally combined with grain boundary pinning strategies for the long-term stabilization of the obtained morphology[5].

## Methods

**Sample fabrication for optical and TEM measurements.** The sample with 24 particles for optical measurements and samples for TEM characterization were fabricated on square, 150 × 150 μm, 40-nm-thick $Si_3N_4$ membranes supported by bulk silicon on all four sides[58]. The two samples with 180 particles each were prepared on an n-doped (100)-Si wafer of 500 μm thickness (SiMat) with 118 nm dry oxide layer grown at 1050 °C in a Centrotherm furnace. The subsequent processing steps were the same for both types of substrates. A double-layer resist mask of 70 nm MMA(8.5)MMA (5 min baking at 180 °C, Microlithography Chemicals Corp.) and 60 nm 950k-PMMA (5 min baking at 180 °C, Microlithography Chemicals Corp.) was spin-coated on the substrate. The nanostructures were defined in the resist mask with a JEOL JBX 9300FS electron-beam system at a dose of 1700 μC/cm² and a beam-step size of 1 nm. The masks were developed for 1 min in a mixed solution of methyl isobutyl ketone (MIBK) and isopropyl alcohol (IPA) with relative concentration of 1:3 and descummed for 3 s with an $O_2$ plasma at 50 W, 250 mTorr (Plasma Therm Batchtop RIE 95 m). Nanoparticles were formed by electron-beam deposition of 30 nm Pd layer through the mask (evaporation rate 1 Å/s in a Lesker PVD 225 Evaporator, base pressure < 5×10⁻⁷ Torr) and lift-off in acetone. The particles were arranged in columns containing 24 or 180 nanodisks of 200 nm in diameter with interparticle distance in the column of 3.5 and 2.5 μm, respectively. Samples for optical and TEM measurements were used without prior annealing.

**Fabrication of samples for GIXRD characterization.** Samples for GIXRD characterization were fabricated using hole-mask colloidal lithography[59], which yields high particle coverage on a surface area of 1 cm² required for GIXRD measurements. The Pd nanodisks were fabricated on square, 1×1 cm silicon substrate by spin-coating 950k-PMMA (5 min baking at 170 °C, Microlithography Chemicals Corp.). The substrate with the PMMA layer was descummed shortly (5 s) in oxygen plasma (50 W, 250 mTorr, Plasma Therm Batchtop RIE 95 m) and water-suspended positively charged polyelectrolyte (poly diallyldimethylammonium (PDDA) MW = 200,000–350,000, Sigma Aldrich, 0.2 wt % in Milli-Q water, Millipore) was dispersed on the surface. Polystyrene (PS) sphere (0.2 wt %, sulfate latex, Interfacial Dynamics Corporation, size 147 nm) solution, which determines the nominal size of the resulting nanodisks, was then deposited on the surface. A thin Cr mask layer was evaporated to cover the PS particles (base pressure of 5 × 10⁻⁷ Torr, evaporation rate 1 Å/s in a Lesker PVD 225 Evaporator). After mask deposition, the PS particles were "stripped" away with tape (SWT-10, Nitto Scandinavia AB), leaving nanoholes in the plasma-resistant film layer. Reactive oxygen plasma etching (5 min at 50 W, 250 mTorr, Plasma Therm Batchtop RIE 95 m) was applied to selectively remove the exposed PMMA layer below the holes. Pd was then evaporated through the hole-mask with thickness of 30 nm to form Pd nanodisks (evaporation rate 1 Å/s in a Lesker PVD 225 Evaporator, base pressure <5 × 10⁻⁷ Torr). Finally, this was followed by a lift-off step in acetone. The samples for XRD characterization were used without prior thermal annealing.

**Single particle kinetics measurements.** For the hydrogen sorption experiments the sample with freshly-deposited Pd was placed in a temperature-controlled, vacuum tight microscope chamber (custom-made Linkam 350 V), which was positioned under an upright optical microscope (Nikon Eclipse LV100, Nikon 50× BD objective) equipped with a motorized stage (Märzhäuser). Schematics of the vacuum setup are depicted in Supplementary Fig. 2. The chamber was connected to a gas inlet supplying 100% $H_2$ gas (6.0 purity) via mass flow controllers (Bronkhorst). For hydrogen absorption measurements the system was initially evacuated to ~10⁻³ mbar pressure and the sample was measured for 5 min in vacuum. After that, the gas inlet was opened to let in at least ~150 to 200 mbar $H_2$ and the system was allowed to saturate for 5 to 10 min. For hydrogen desorption measurements, the outlet of the stage was connected to a vacuum pump (Pfeiffer, HiPace 80) via a pneumatic valve. After the sample was saturated with hydrogen, the pneumatic valve was opened in order to quickly pump the chamber back to initial ~10⁻³ mbar pressure and the measurement was recorded for another 5 to 15 min. For such kinetics experiments an array of particles was aligned in the field of view of the microscope (Fig. 1) and their image was captured along the duration of the experiment with a thermoelectrically cooled Electron Multiplying Charge Coupled Device (EMCCD) camera Andor iXon Ultra 888. The scan rate for image acquisition varied between 10 to 1 frame(s) per second. The intensity of each particle was then analyzed as the sum of brightest pixels constituting the diffraction limited spot of light scattering from each particle in the acquired images. The temperature range of experiments was between 303 to 338 K with 5 K steps. The $T$-order at which particles were measured for the mixed $T$-sweeps is shown in Supplementary Fig. 4. For each of the series of measurements with consistent $T$-sweep up or down,

a fresh sample was used. All three samples prior to beginning of the measurements for extraction of Arrhenius parameters were pre-cycled 3 times at 303 K. The activation energy was extracted from each set of measurements containing the full $T$ range 303–338 K.

**Single particle isotherm measurements.** The isotherm measurements were performed in the same setup as kinetics experiments with the exception that the vacuum pump was removed from the gas outlet. The measurement chamber was then connected to a set of mass flow controllers (Bronkhorst, Low-$\Delta$P-flow and EL-flow) to supply the desired gas flow and concentration to the sample. We used Ar as carrier gas (6.0 purity) and mixed it at different concentrations with 100% $H_2$ gas (6.0 purity) and operated the system at atmospheric pressure. CCD images of the particles were captured along the duration of the experiment using 10 s acquisition time for each image. The $H_2$ absorption and desorption isotherms at 303 K were measured on the sample used for $T$-mixed kinetic sweeps and on a freshly prepared sample with 24 Pd nanoparticles without previous exposure to hydrogen.

**TEM characterization.** The samples prepared on TEM membranes (analogous to the ones measured in optical experiments) were imaged as deposited and after 12 and 30 (de)hydrogenation cycles in a FEI Titan 80-300 (LaB6 filament, operated at 300 kV). Imaging was done in bright field-mode at 87k× magnification.

**GIXRD measurements.** Grazing Incidence X-ray Diffraction was used to determine the crystallite size. The measurements were carried out using a Mat:Nordic SAXSLAB benchtop beamline equipped with a Rigaku 003 microfocusing Cu X-ray source producing a parallel beam, and two Dectris detectors: Pilatus3 300 K R (orthogonal to the beam) and 100 K (on a goniometer circle centered at the sample position). The entire beam path was evacuated during measurement to reduce air scattering and increase signal quality. Instrumental broadening was determined before each measurement using Corundum powder. The incidence angle was set at 2° in order to maximize the signal from the sample and peak shape. The acquired 2D images were reduced using standard protocols with the SAXSGUI software. The FWHM of the Pd 111 reflection at 40° 2θ was determined using a combination of Gaussian and Lorentzian functions, and the instrumental contribution was subtracted to obtain the peak widening from crystallite size effects. The Scherrer equation was then used to estimate the crystallite sizes. To determine crystallite sizes after number of cycles ($N$), every time a new sample cycled $N$ times was used. Exposure time for each measured nanodisk sample was 0.5 to 2 h.

## Data availability
All relevant data sets generated during and/or analyzed during the current study are available from the corresponding author upon request.

## Code availability
All relevant code written to control the experimental setup and analyze the scattering intensity data are available from the corresponding author upon request.

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

## Acknowledgements
This study has received funding from the European Research Council (ERC) under the European Union's Horizon 2020 research and innovation program (678941/SINCAT) and from the Knut and Alice Wallenberg Foundation projects 2015.0055 and 2016.0210. Part of this work was carried out at the MC2 cleanroom facility and at the Chalmers Materials Analysis Laboratory. V.P.Z. was partly supported by the Ministry of Science and Higher Education of the Russian Federation in the framework of the budget project for Boreskov Institute of Catalysis (grant 0239-2021-0011). We also acknowledge fruitful discussion with Profs. H. Grönbeck and A. Hellman.

## Author contributions
S.A. executed all single particle experiments, analyzed the data and co-wrote the manuscript; M.S. executed the GIXRD experiments and analyzed the data; S.N. executed the TEM analysis, J.F. nanofabricated the samples; V.P.Z. contributed the theoretical analysis of the kinetics and co-wrote the manuscript; C.L. co-wrote the manuscript, supervised S.A. and coordinated the project as a whole.

## Funding

## Competing interests
The authors declare no competing interests.
