## [Peer Review File · Nature Communications]

Grain-Growth Mediated Hydrogen Sorption Kinetics and Compensation Effect in Single Pd NanoparticlesREVIEWER COMMENTS

Reviewer #1 (Remarks to the Author):

The authors study the kinetics of hydrogen absorption and desorption in individual polycrystalline Pd disks with diameters of 200 nm. The single-particle absorption and desorption kinetics significantly decrease upon cycling, which also induces grain coarsening. The grain coarsening is also indicated as the origin of the compensation effect between activation energy and the pre-exponential factor t_0 . The data is impressive in its quality and quantity. The first part of the paper is clearly written and organised, but I found the analysis of compensation effects to be very difficult to follow. Furthermore, despite acquiring an impressive amount of data, the authors have sometimes missed the opportunity to extract quantitative information from them (see, for example, my major comments 3, 6, and 8). Given the above considerations and the major comments below, I believe that publication of the paper in its present form would be premature. I hope my comments can help improving the manuscript.

Major comments

1) I find the following sentence a bit too strong: "corresponding kinetic studies with single nanoparticle resolution do not exist since in the few cases where the temporal evolution of the transition from one state to another was resolved,^{23,24} it was neither analysed nor quantified".

In figure 5 of reference 24, the authors plot the phase front position as a function of time for single particles of different shapes and fit the trends with linear functions. From the linear trends observed they conclude that "the reaction cannot be limited by the diffusion of hydrogen atoms into the bulk of the particle, which would follow a \sqrt{t} curve". Wouldn't this qualify as an analysis and quantification of their data?

2) Figure 1d, 1st absorption cycle: is there a reason for the lower number of measurements with respect to the 2nd and 3rd absorption cycles (which appear to have at least twice as many data points, i.e. roughly 2 per second)? Also, while the position of the red circles appears logical for all other panels, for the 1st absorption it appears to be misplaced, namely well before (in time) the transition to the hydride state. In fact, a very similar t_{50} for all particles for the first absorption cycle is acknowledged by the authors when they write "Moreover, the time constant for the very first hydrogen absorption is essentially identical for all 24 particles but starts to scatter more between them already for the first desorption step, hinting at a hydrogen-induced evolution of increasingly more single-particle specific response upon repeated cycling (Fig. 1d)". I am not entirely convinced by the data (the red circle is not in the middle of the "jump" and there is only 1 datapoint per second which limits the time resolution) or the explanation (I don't see why the single particle character should increase with cycling and it is perhaps more reasonable to assume that the 1st transition was too fast to be caught with a 1 frame per second rate?).

3) "Theory predicts $m = 2$, whereas experiments usually show m in the range from 3 to 5. Fitting eq. (2) to our GIXRD data yields very good agreement for $m = 5$ (Fig. 3b)". What about the parameter A^* ? What is its fitted value and how does it compare with theoretical predictions? Also, what does an m value of 5 mean in terms of the kinetics of the process? Why is it common for macroscopic samples? Why is the theory so wrong?

4) "Strikingly, for absorption, the average t_{50} increases from 1.6 to 10.9 s, and for desorption from 13.4 to 83.7 s". This is indeed a striking difference in behavior, underlined by the change in crystallite size upon cycling presented in Figure 3. Doesn't this "aging" behavior undermine the validity of the Arrhenius analysis of Figure 2? If the absorption and desorption processes slow down with cycling, how can you extract thermodynamic parameters from the temperature sweep procedure given in SI Fig. 4? Wouldn't you be seeing the convoluted effect of changing the temperature AND changing the sample morphology (and therefore kinetic properties) due to cycling? This aspect is then tackled extensively when discussing compensation effects, but I don't see how it can be mitigated by using different temperature sweeps: the grain coarsening with cycling will always happen, regardless of the direction of the T-sweep.

5) "Furthermore, we notice a significant broadening in the t_{50} distribution for increasing number of cycles, which indicates a dramatically increased particle individuality imposed by the single particle specific evolution of the grain structure". I am not sure I understand what conclusion can be derived by this observation. For the first cycle, the broadening is limited by the speed of acquisition (see my 2nd comment above), for subsequent cycles I would relate the broadening to the mean value: is the broadening a larger percentage of the mean? If so, I could see the value of pointing out this behavior, but then I would also add a sentence speculating why this is happening.

6) "In addition, the observed structural evolution not only is reflected in the single particle sorption kinetics, but also in the corresponding thermodynamics, where single particle pressure-composition isotherms reveal significant increase in hysteresis upon cycling (SI Fig. 11), thereby corroborating recent results correlating grain boundary length with hysteresis width". Are the isotherms for the 1st and the 41st cycles measured at the same rate of pressure increase? If so, the observed larger hysteresis loops for the 41st cycle could be due to the slowing dynamics of hydrogen absorption and desorption. In essence, how can you be sure that the observed effect is thermodynamic in nature and not kinetic? Furthermore, the authors quantify the hysteresis loop widening and mention that this is correlated to grain boundary length, but fail to relate the two quantitatively: should this remain just an observation or can we extract structural information from the measured absorption and desorption pressures before and after cycling? Is the magnitude of widening reasonable given the measured coarsening of the grains given in Figure 3b?

7) "Finally, from an application perspective, the found evolution of structure and kinetics over time both identifies a potentially critical feature responsible for the deterioration of (Pd-based) solid-state hydrogen storage materials and hydrogen sensors over time and a so far unexplored design rule for the development of ultrafast hydrogen sensors in par with the US DoE response time target, namely stabilization of grain structure". I think that the word "both" is misplaced. But most importantly, I find this claim to be exaggerated. Grain coarsening in palladium upon hydrogen cycling and its relation to a slowing down of the absorption and desorption kinetics are not new results. The relationship between density of grain boundaries and hydrogen transport in palladium is well established, for example in the context of hydrogen separation membranes (see for example <https://www.sciencedirect.com/science/article/abs/pii/S0921509395099506>) and the effect can be mitigated by alloying with different metals. The sentence in the abstract "This constitutes the first observation of the impact of grain growth on kinetic processes occurring inside nanostructured materials" is also an unnecessary overstatement.

8) "The difference between these energies, ≈ 0.14 eV, is significant and indicates that grain boundary diffusion is fast, and indeed may account for the observed dramatically increasing t_{50} values upon grain growth". Again I find this qualitative analysis to be incomplete. The authors have measured the average crystallite size and the absorption and desorption kinetics before and after cycling, know the dimensions of the disks, and have an estimate of the difference in activation energy for hydrogen diffusion at grain boundaries. Can they not relate these values to each other? Does the measured slowing kinetics make sense given the measured coarsening? This kind of analysis would really improve the quality of the paper. Similarly, in SI section 12 the authors write "the faster desorption kinetics we observe in the small grain regime can be understood from an energetics point of view, where it is known that the activation energy for H₂ desorption from more open Pd(211) and Pd(100) faces is lower than from Pd(111), with a computed energy difference of about 0.1 eV". How does this value of 0.1 eV relate to the measured slower desorption kinetics between the first and the subsequent cycles?

9) The analysis of compensation effects is very detailed but also extremely hard to follow and I would recommend the authors to make an effort to improve the clarity of their text, make it accessible to a typical reader of Nature Communications (i.e. not an expert in compensation effects), and reduce the current fragmentation between main text and SI, which leaves the reader very confused. An example of the level of complexity that is hard to follow is in the insets of Fig. 5a-c, but in general it is unclear to me how the authors can conclude "we have unambiguously identified the grain-growth process as the

mechanistic origin of the observed kinetic compensation effect".

10)"Looking forward, our findings advocate wide opportunities for the practical application of kinetic compensation effects for the screening and tailoring of temperature or other technically important parameters at which chemical transformations occur on nanoparticles and nanostructures". I am not sure what is the meaning of this sentence. I am also puzzled by the final remark "Furthermore, they have direct implications for the fundamental understanding of metal-solute interactions and unravel a so far unexplored handle for the rational design of nanomaterials that find application in solid-state hydrogen storage and hydrogen sensors with ultrafast response and long-term stability. Specifically, it becomes clear that single crystalline structures obtained, for example, by colloidal synthesis very likely constitute inferior solutions and corresponding development should be steered towards highly polycrystalline structures, ideally combined with grain boundary pinning strategies for the long-term stabilization of the obtained morphology". Inferior solutions to what problem? Under which performance specifications? For example, one could argue that colloidal synthesis is much cheaper than electron-beam deposition...

Minor comments

"electron-beam lithography onto an oxidized silicon wafer (Fig. 1a)". Figure 1a, however, shows transmission electron images: how thick was the silicon wafer? Or are these deposited on a TEM substrate? If so, it should be mentioned. Also, do you expect a different grain structure when depositing on different substrates?

Typo: "and ensembles of Pd nanodisks³⁸ and"

The determination of t_{50} crucially depends on the speed at which the pressure changes in the sample chamber. While a reference to the previous publication explaining the setup is given ("To resolve the fast hydrogen sorption kinetics for all single particles simultaneously, we have employed multiplexed single-particle plasmonic nanoimaging microscopy³⁴"), it would be desirable to mention this aspect also here.

A discussion of the meaning and significance of the "sum of square errors (SSE), R-square, adjusted R-square and Root Mean Squared Error (RMSE)" in SI Fig. 10 would be nice. What do these values mean?

"Deriving the average crystallite size after each step reveals": an explanation of how GIXRD patterns are analysed to extract the crystallite size should be given, or if an explanation would be too trivial, at the very least a reference.

Why also fitting the cycling evolution of the crystallite size with a logarithmic function, if "the conventional power-law representation for the average radius of a crystallite during grain growth reads as" equation (2)? What do we learn from the logarithmic fit?

"To further corroborate the GIXRD results, we performed transmission electron microscopy (TEM) characterization of a single nanoparticle before cycling, and after 12 and 30 (de)hydrogenation cycles, respectively, which confirms the significant hydrogen sorption-induced grain growth (Fig. 3c)": is this the same particle? Also, why showing both 12 and 30 cycles? They don't seem to really differ in the grain size, as also expected from the trend in Figure 3b.

The histograms in Figure 4a and 4b are very difficult to read. For example, it took me a while to realize that absorption cycles 2 and 3 are overlapping. I suggest plotting the histogram separately in the SI and replace them here with the average and standard deviation for each cycle.

Typo: "by taking found the grain-growth"

Reviewer #2 (Remarks to the Author):

This communication reports on a systematic study of the hydriding kinetics of Pd nanoparticles, with plasmonic microscopy as the main tool. This approach enables tracking the hydriding kinetics of single particles while correlating it with the evolution of their microstructures. The results are truly impressive, in that they provide a quantitative picture of the evolution of both crystallite configuration and hydriding kinetics upon multiple H₂ cycling, and establishes a convincing link between them. Therefore, I think that this paper is suitable for publication in Nature Communications. However, I have some comments which should be addressed, notably concerning the concentration of hydrogen at defect sites under vacuum and the decoupling of grain boundary effects from these of other defects.

Important general comments:

1) The rapidly slowing kinetics observed by the authors during H₂ absorption and desorption is not something that is systematically observed on similar Pd-based nanomaterials, to the best of my knowledge (I am thinking about thin Pd films in particular). While wondering why this could be the case here, I realized that one important experimental aspect of the kinetic study is not discussed at all in the paper: H₂ is simply vacuumed and not exposed to air between the absorption/desorption cycles. Re-exposing the particles to H₂ after vacuuming or venting are two very different situations. Indeed, if one looks for example at Phys. Chem. Chem. Phys. 2011 (13) 11412 (Fig. 2), it was shown for thin Pd films that residual hydrogen remains in the films after vacuuming, notably because part of the H atoms remain trapped in crystalline defects, which are deep potential wells (but venting will always result in a full hydrogen desorption). Obviously, if this is the case, the kinetics of the first absorption will be very different from the second one and could explain what appears as a “rapidly slowing” kinetics from cycle to cycle. Furthermore, if grain boundaries are evolving between each cycle, so will the amount of trapped hydrogen under vacuum.

I don't contest the link that the authors establish between the evolution of crystallite sizes and the kinetics of H₂ absorption/desorption, but I think my argument should be addressed by the authors and an assumption on the concentration of hydrogen after vacuuming in each absorption/desorption cycle should at least be made, e.g. in the experimental part.

2) The increase of Pd crystallite size with the number of absorption/desorption cycles is also not something that is commonly observed, at least with bigger crystallites (e.g. from several tens of nanometers like in thin films). I guess this is only happening because the defect density is very high here. Do the authors expect defect recovery to reach a plateau? From the fit in Fig. 3b for example, do the authors identify a value at which the crystallite size saturates? If so, this could be a valuable information to add to the paper.

3) The effect of crystallite size on the kinetics of H₂ absorption/desorption seems clear here, but I am not comfortable with the absence of related discussion on the effect of other crystalline defects. I am sure the authors are aware that other crystalline defects (dislocations, vacancies, twins, impurities, etc) also affect the hydriding mechanism (see e.g. A. Pundt and R. Kirchheim, Annu Rev Mater Res 36 (2006) 555). In the best case scenario, the other defects are not evolving with H₂ cycling, but this is most likely not the case (e.g. in Int. J. Hydrogen Energy 40 (2015) 7335, thin films absorbing hydrogen, even in the alpha-phase, show a significant increase in dislocation density). This is a very complex situation: the density of some defects might increase, decrease or stay constant. First of all, the authors should make assumptions on the behavior of other defects with H₂ cycling, and second, they should explain why they think that the effect of grain boundaries is the main effect studied here. And of course, any experimental evidence to support these assumptions (e.g. defect statistics from ex-situ characterization) are highly valuable.

Specific comments:

1) Page 2, line 5: I don't like the term “completely uncharted” here. On page 3 line 15, the authors use the term “widely lacking” which seems more appropriate to me with respect to the state of the art that the authors correctly describe in the introduction.

2) Page 4, line 22: The crystalline orientation of the oxidized silicon wafers should be specified.

Point-to-Point Response, Alekseeva et al.

Report of Referee 1

The authors study the kinetics of hydrogen absorption and desorption in individual polycrystalline Pd disks with diameters of 200 nm. The single-particle absorption and desorption kinetics significantly decrease upon cycling, which also induces grain coarsening. The grain coarsening is also indicated as the origin of the compensation effect between activation energy and the pre-exponential factor t_0 . The data is impressive in its quality and quantity. The first part of the paper is clearly written and organised, but I found the analysis of compensation effects to be very difficult to follow. Furthermore, despite acquiring an impressive amount of data, the authors have sometimes missed the opportunity to extract quantitative information from them (see, for example, my major comments 3, 6, and 8). Given the above considerations and the major comments below, I believe that publication of the paper in its present form would be premature. I hope my comments can help improving the manuscript

Our reply: We thank the Referee for the positive assessment of the data quality and quantity in our work and the large number of insightful and constructive specific comments given. We address them in detail below.

Major comments

1) I find the following sentence a bit too strong: "corresponding kinetic studies with single nanoparticle resolution do not exist since in the few cases where the temporal evolution of the transition from one state to another was resolved,^{23,24} it was neither analysed nor quantified". In Figure 5 of reference 24, the authors plot the phase front position as a function of time for single particles of different shapes and fit the trends with linear functions. From the linear trends observed they conclude that "the reaction cannot be limited by the diffusion of hydrogen atoms into the bulk of the particle, which would follow a \sqrt{t} curve". Wouldn't this qualify as an analysis and quantification of their data?

Our reply: We agree with the Referee's opinion that our remark is too categorical. We have reworded to:

"However, corresponding *kinetic* studies with single nanoparticle resolution are very scarce and in the few cases where the temporal evolution of the transition from one state to another was resolved,^{23,24} it was not quantitatively analyzed with respect to, e.g., the kinetic parameters."

2) Figure 1d, 1st absorption cycle: is there a reason for the lower number of measurements with respect to the 2nd and 3rd absorption cycles (which appear to have at least twice as many data points, i.e. roughly 2 per second)? Also, while the position of the red circles appears logical for all other panels, for the 1st absorption it appears to be misplaced, namely well before (in time) the transition to the hydride state. In fact, a very similar t_{50} for all particles for the first absorption cycle is acknowledged by the authors when they write "Moreover, the time constant for the very first hydrogen absorption is essentially identical for all 24 particles but starts to scatter more between them already for the first desorption step, hinting at a hydrogen-induced evolution of increasingly more single-particle specific response upon repeated cycling (Fig. 1d)". I am not entirely convinced by the data (the red circle is not in the

middle of the "jump" and there is only 1 datapoint per second which limits the time resolution) or the explanation (I don't see why the single particle character should increase with cycling and it is perhaps more reasonable to assume that the 1st transition was too fast to be caught with a 1 frame per second rate?).

Our reply: For the 1st cycle, indeed the measurement rate was 1 frame per second - that was just how we happened to measure the first cycle for this sample. For all other cycles and also other two samples (with temperature sweep directions T_{up} and T_{down}) the measurement rate was 2 frames per second, and we have added corresponding figures below for comparison of first cycle measurements between the three samples. Clearly, the time-evolution of the response is very similar for all three cases, irrespective of the frame rate and thus the almost identical response from all particles is also not a consequence thereof.

When it comes to or explanation of the effect, a potentially more important factor than the frame rate is the time constant of the used measurement chamber with respect to gas exchange. We discuss this aspect in detail in response to Comment # 5 below, with the key point being that our instrument has a time constant on the order of 0.5 s. This value is significantly below the temporal response of the samples even at the first hydrogenation but it is at the same time also reasonably close. Therefore, it is indeed not possible to, at least partially, rigorously exclude some convolution between the intrinsic individual particle response and the instrument time constant for the first hydrogenation cycle as (part of) the reason for the almost identical response for all particles in the first hydrogenation cycle. To clarify this point, we have added the following sentence to the revised manuscript:

“At the same time, we also note that the time constant of our instrument and the temporal response of the particles during the first hydrogenation cycle ($t_{50} \approx 1.5$ s) are close enough that we cannot completely exclude some convolution of these two factors and this being a contributing factor for the observed identical response.”

As the next point, we thank the reviewer for pointing out the oddity in Fig. 1d, where indeed the pixels were slightly displaced, now we have corrected it and there is a new version of Figure 1 in the main text and pasted below for convenience:

3) "Theory predicts $m = 2$, whereas experiments usually show m in the range from 3 to 5. Fitting eq. (2) to our GIXRD data yields very good agreement for $m = 5$ (Fig. 3b)". What about the parameter A^* ? What is its fitted value and how does it compare with theoretical predictions? Also, what does an m value of 5 mean in terms of the kinetics of the process? Why is it common for macroscopic samples? Why is the theory so wrong?

Our reply: To answer the Referee's questions and address them in the revised manuscript, we have re-written and expanded the quoted section to:

"The power law, represented in terms of the average radius of a crystallite during grain growth, Eq. (2), where $\langle R_{cr} \rangle_0$ is the initial average crystallite size, and A^* and m are parameters, has been widely used in studies focused on grain growth in macroscopic samples.⁵ To this end, the simplest theory⁴⁰ focused exclusively on the interface curvature as a driving force for grain growth predicts $m = 2$, whereas experiments and more advanced models⁵ usually show m in the range from 3 to 5 (sometimes up to 12, which then means that the growth is nearly terminated). This wide range of observed m values is the consequence of that the exact kinetics of grain growth depends on a multitude of factors in the material (e.g., triple junctions, non-equilibrium vacancies and nanovoids, and complex diffusion of atoms)⁵, and also reflects the fact that a detailed full-scale interpretation of this process and the exact mechanistic meaning of experimentally derived values of A^* and m is still lacking. Fitting eq. (2) to our GIXRD data yields good agreement for $m = 5$ (Fig. 3b), a value common also in macroscopic nanocrystalline samples.⁵ For our case at hand, this means that we indeed are observing a grain-growth process with kinetic characteristics common for nanocrystalline materials. "

4) "Strikingly, for absorption, the average t_{50} increases from 1.6 to 10.9 s, and for desorption from 13.4 to 83.7 s". This is indeed a striking difference in behavior, underlined by the change in crystallite size upon cycling presented in Figure 3. Doesn't this "aging" behavior undermine the validity of the Arrhenius analysis of Figure 2? If the absorption and desorption processes slow down with cycling, how can you extract thermodynamic parameters from the temperature sweep procedure given in SI Fig. 4? Wouldn't you be seeing the convoluted effect of changing

the temperature AND changing the sample morphology (and therefore kinetic properties) due to cycling? This aspect is then tackled extensively when discussing compensation effects, but I don't see how it can be mitigated by using different temperature sweeps: the grain coarsening with cycling will always happen, regardless of the direction of the T-sweep.

Our reply: In essence, the Referee here indeed summarizes a key message of our paper, i.e., that such ageing effects likely convolute an Arrhenius analysis and therefore are the reason for the observed compensation effects. Concerning this aspect, we note that except the simplest one-step processes described by the mass-action law, an Arrhenius analysis is always "phenomenological" and therefore by itself does not need validation. This is reflected in the fact that the corresponding Arrhenius parameters are considered to be "apparent" (this widely used term is employed in our article as well). With this reservation, we can add that what needs validation is the *interpretation* of the apparent Arrhenius parameters obtained by using this analysis, which is exactly what we do in detail in this work.

To make this even more clear, we have added the word “phenomenologically” above equation 1 on page seven, such that the corresponding sentence now reads as:

“From this data set, we then extracted the corresponding apparent activation energies for each particle, by assuming that t_{50} can phenomenologically be represented in the Arrhenius form by...”

Furthermore, we added the following sentence to the conclusions section on page 19 of the revised manuscript:

“In other words, we have observed convolution between the conventional Arrhenius effect of temperature and the effect of grain growth on the kinetics.”

Finally, to comment on the Referee’s last point, indeed, the grain coarsening with cycling does always happen regardless of the direction of the T -sweeps. Its *relative* effect at low and high temperatures is, however, different depending on the direction of the T -sweeps. In other words, as we explain in the manuscript, the direction does matter for the result obtained from Arrhenius analysis. To also explicitly discuss this point here, we add the following general remark: Let us consider that the process rate is measured within the temperature range from T_1 to T_2 ($T_1 < T_2$). The Arrhenius parameters depend on the ratio of the process rates at these temperatures. If the process is simple and its rate is determined only by temperature, the direction of the variation of temperature does not matter. If there is another irreversible process (e.g., the grain growth as in our case) and with increasing time it results in decrease of the rate of the main process irrespective of the direction of the variation of temperature, the ratio the main-process rates at T_1 and T_2 measured with increasing T from T_1 to T_2 will be decreased by the second process (because the measurement at $T = T_2$ is performed later on), whereas this ratio measured with decreasing T will be decreased by the second process (because the measurement at $T = T_1$ is performed later on). Thus, the direction does matter.

5) "Furthermore, we notice a significant broadening in the t_{50} distribution for increasing number of cycles, which indicates a dramatically increased particle individuality imposed by the single particle specific evolution of the grain structure". I am not sure I understand what conclusion can be derived by this observation. For the first cycle, the broadening is limited by the speed of acquisition (see my 2nd comment above), for subsequent cycles I would relate the broadening to the mean value: is the broadening a larger percentage of the mean? If so, I could

see the value of pointing out this behavior, but then I would also add a sentence speculating why this is happening.

Our reply: To accommodate the Referee's comment, we have added the following text to the revised manuscript on p.11:

“The key factor here is that with increasing number of cycles the average number of grains becomes much smaller. The measure of the effect of this factor on t_{50} is the ratio of the width of the distribution of this number to the average number. According to the Poisson distribution, this ratio increases with decreasing the average number of grains. From this perspective, the system becomes more heterogeneous with increasing number of cycles, and it results in a significant broadening in the t_{50} distribution.”

6) "In addition, the observed structural evolution not only is reflected in the single particle sorption kinetics, but also in the corresponding thermodynamics, where single particle pressure-composition isotherms reveal significant increase in hysteresis upon cycling (SI Fig. 11), thereby corroborating recent results correlating grain boundary length with hysteresis width". Are the isotherms for the 1st and the 41st cycles measured at the same rate of pressure increase? If so, the observed larger hysteresis loops for the 41st cycle could be due to the slowing dynamics of hydrogen absorption and desorption. In essence, how can you be sure that the observed effect is thermodynamic in nature and not kinetic? Furthermore, the authors quantify the hysteresis loop widening and mention that this is correlated to grain boundary length, but fail to relate the two quantitatively: should this remain just an observation or can we extract structural information from the measured absorption and desorption pressures before and after cycling? Is the magnitude of widening reasonable given the measured coarsening of the grains given in Figure 3b?

The isotherms are measured in gas flow mode (i.e. at atmospheric pressure), not in the vacuum conditions. The hydrogen partial pressure is increased/decreased in step-wise fashion during the measurements, and at each pressure step there is dwelling time in order to allow for all the particles to reach a new steady-state at the corresponding pressure before moving on to the next pressure step. The samples that were used in the isotherm measurements were: a sample in as-deposited condition (0 cycles, i.e., 0 hydrogen exposures) and a sample already used in kinetics measurements (T_{mix} , i.e. after 41 cycles). For each of these samples we used the same measurement script with the same pressure steps and dwelling times at each of the steps. We have added corresponding isotherms for each of the particles to the SI Fig. 11 and have clarified the text in SI section 9.

Below is an example of a measurement from one of the particles: raw data of the scattering intensity trace over time, and where each red cross indicates the end of the hydrogen partial pressure step.

To answer the question about the magnitude of the hysteresis widening: yes, it is in line with our previous observations for *annealed* Pd nanoparticles (Alekseeva, S. *et al. Nature Comm.* **8**, 1084 (2017)), where hydride formation pressure was shown to be correlated with grain boundary length. Specifically, we see (figure below) that the hysteresis width in the present study is smaller than in the previous one, where we had used thermally annealed Pd particles. The reason is indeed that upon annealing, the grain size increases from sub-10 nm size in as-deposited condition up to the size of the particle itself (below we show results measured from annealed Pd particles with diameter of 150 nm). The number of grains then depends on annealing conditions, but with the ones for the sample used for the comparison here (12 h at 400 °C), we can get single grain particles, as well as particles with up to 2-10 grains. As expected, comparing the present system with the annealed one, we see that hysteresis for as-deposited particles is significantly narrower than for annealed ones, and the hydrogen-sorption induced cycling increases hysteresis width somewhat but to a level that still is significantly below the annealed particles with few grain boundaries only.

7) “Finally, from an application perspective, the found evolution of structure and kinetics over time both identifies a potentially critical feature responsible for the deterioration of (Pd-based) solid-state hydrogen storage materials and hydrogen sensors over time and a so far unexplored design rule for the development of ultrafast hydrogen sensors in par with the US DoE response time target, namely stabilization of grain structure”. I think that the word “both” is misplaced. But most importantly, I find this claim to be exaggerated. Grain coarsening in palladium upon

hydrogen cycling and its relation to a slowing down of the absorption and desorption kinetics are not new results. The relationship between density of grain boundaries and hydrogen transport in palladium is well established, for example in the context of hydrogen separation membranes (see for example <https://www.sciencedirect.com/science/article/abs/pii/S0921509395099506>) and the effect can be mitigated by alloying with different metals. The sentence in the abstract “This constitutes the first observation of the impact of grain growth on kinetic processes occurring inside nanostructured materials” is also an unnecessary over-statement.

Our reply: We have deleted the word “both” in the revised manuscript. When it comes to our claim, we agree that “the relationship between density of grain boundaries and hydrogen transport in palladium is well established” is correct in the context of *macroscopic* samples. To the best of our knowledge, however, such relationships have not been studied and reported for *nanoparticles* and are therefore indeed new. To make this explicitly clear, we have reworded the sentence in the abstract to:

“This constitutes the first observation of the impact of grain growth on kinetic processes occurring inside nanoparticles, provides a physically sound mechanistic explanation of kinetic compensation effects, and highlights the importance of single particle experiments for their systematic investigation.”

And in the conclusions section to:

“Finally, from an application perspective, the found evolution of structure and kinetics over time identifies a potentially critical feature responsible for the temporal evolution of the structure of (Pd-based) nanoparticles used in hydrogen storage systems, and a so far unexplored design rule for the development of ultrafast hydrogen sensors in par with the US DoE response time target^{18,41}, namely the optimization and stabilization of grain structure.”

8) *“The difference between these energies, ≈ 0.14 eV, is significant and indicates that grain boundary diffusion is fast, and indeed may account for the observed dramatically increasing t_{50} values upon grain growth”. Again, I find this qualitative analysis to be incomplete. The authors have measured the average crystallite size and the absorption and desorption kinetics before and after cycling, know the dimensions of the disks, and have an estimate of the difference in activation energy for hydrogen diffusion at grain boundaries. Can they not relate these values to each other? Does the measured slowing kinetics make sense given the measured coarsening? This kind of analysis would really improve the quality of the paper. Similarly, in SI section 12 the authors write “the faster desorption kinetics we observe in the small grain regime can be understood from an energetics point of view, where it is known that the activation energy for H₂ desorption from more open Pd(211) and Pd(100) faces is lower than from Pd(111), with a computed energy difference of about 0.1 eV”. How does this value of 0.1 eV relate to the measured slower desorption kinetics between the first and the subsequent cycles?*

Our reply: Here we do not agree with the Referee in that we are not trying “hard enough” to quantitatively analyze and relate available information. We indeed establish clear relations and they indeed are self-consistent from our point of view. The key problem here is that the processes we are observing and attempt to quantitatively analyze are very complex and therefore cannot be interpreted in all the details that one could wish for. In fact, we find it very important to not overinterpret our data by attempting to draw too many and too rigorous

quantitative conclusions. In the specific case at hand, it is important to be aware of that, as we explicitly explain in the manuscript, the energy difference of 0.14 eV is a coarse *estimate* that we base on the assumption that grain boundaries can be thought of as a (111) surface. This is of course a significant simplification that therefore shall not be used to calculate any explicit kinetics. At the same time, as we do here, it is helpful to give a rough estimate of the likely energy landscape at hand, where the main point is that it indeed is reasonable to expect that E_a at a grain boundary is lower than for diffusion through a hydride and that therefore diffusion along GBs is expected to be faster than through the grains. The exact magnitude of the involved barriers is unknown and therefore, no quantitative conclusions should be drawn from our estimation.

Regarding the second point, i.e., if “the measured slowing kinetics make sense given the measured coarsening”, it is exactly the point of the mentioned estimated activation energies to show that they do. A more quantitative analysis is not sensible as outlined above.

Regarding the third point, we note that growth of grains is the fastest during the first few hydrogenation cycles [Eq. (2)]. In addition, this initial grain structure is far from equilibrium (as a consequence of the nanofabrication/evaporation process) and rapidly evolves during these cycles. Physically, the energy, 0.1 eV, indicated in our text represents *the scale* of the corresponding change of the binding energy of H atoms, and this value can be used for rough estimates as we do, but not for more detailed quantitative analysis, again for the same reasons as outlined above.

9) The analysis of compensation effects is very detailed but also extremely hard to follow and I would recommend the authors to make an effort to improve the clarity of their text, make it accessible to a typical reader of Nature Communications (i.e., not an expert in compensation effects), and reduce the current fragmentation between main text and SI, which leaves the reader very confused. An example of the level of complexity that is hard to follow is in the insets of Fig. 5a-c, but in general it is unclear to me how the authors can conclude "we have unambiguously identified the grain-growth process as the mechanistic origin of the observed kinetic compensation effect".

We thank the reviewer for the comment regarding the fragmentation between main text and SI. To amend this:

- (i) we have moved the discussion of desorption mechanism from the SI (formerly SI section 12) into the main text and accordingly added a panel in Fig. 4e;
- (ii) we believe that specifically the analysis related to compensation effect according to the model by Griessen et al. [Ref. 30] was "hard to follow" since it is oriented to experts and thus likely not for a more general readership, we have therefore moved the corresponding text into the SI in the revised version;
- (iii) we have modified Fig. 5d-f in the main text to include the illustration of the effect of direction of the T measurements on the Arrhenius analysis, which was formerly discussed in SI section 16, in order to reduce the need for the readers to have to resort to the SI.

10) "Looking forward, our findings advocate wide opportunities for the practical application of kinetic compensation effects for the screening and tailoring of temperature or other technically important parameters at which chemical transformations occur on nanoparticles and nanostructures". I am not sure what is the meaning of this sentence.

I am also puzzled by the final remark "Furthermore, they have direct implications for the fundamental understanding of metal-solute interactions and unravel a so far unexplored handle

for the rational design of nanomaterials that find application in solid-state hydrogen storage and hydrogen sensors with ultrafast response and long-term stability. Specifically, it becomes clear that single crystalline structures obtained, for example, by colloidal synthesis very likely constitute inferior solutions and corresponding development should be steered towards highly polycrystalline structures, ideally combined with grain boundary pinning strategies for the long-term stabilization of the obtained morphology". Inferior solutions to what problem? Under which performance specifications? For example, one could argue that colloidal synthesis is much cheaper than electron-beam deposition...

Our reply: Regarding the first point, we have deleted the sentence that was not clear from the revised manuscript.

Regarding the second point, what we mean is that colloidal nanoparticles may be an inferior solution for both hydrogen storage and hydrogen sensor solutions because they usually are *single crystals*. This means that they will yield slower hydrogen sorption times than polycrystalline systems that we have investigated here since, as we discussed in detail, hydrogen diffusion is more rapid along grain boundaries. To make this clear and, as requested by the Referee, explicitly mention the performance-specification we have in mind, we have slightly reworded to:

*“Furthermore, they have direct implications for the fundamental understanding of metal-solute interactions and unravel a so far unexplored handle for the rational design of nanomaterials that find application in solid-state hydrogen storage and hydrogen sensors with ultrafast response and long-term stability. Specifically, it becomes clear that single crystalline structures obtained, for example, by colloidal synthesis very likely constitute inferior solutions *with respect to hydrogen sorption kinetics* and corresponding development should be steered towards highly polycrystalline structures, ideally combined with grain boundary pinning strategies for the long-term stabilization of the obtained morphology”*

Finally, when it comes to the issue of cost of nanolithography, we note the following two points:

- Nanolithography techniques are the work horse of microelectronics, which likely is one of the most scaled industries there is. It very widely relies on electron-beam and sputter deposition of materials in combination with a myriad of different nanolithography techniques. Therefore, there exist many technologies that enable the very cost-efficient fabrication of polycrystalline nanoparticles on surfaces. We used electron-beam lithography specifically in this study for the purpose of preparing surfaces that are optimized for our experimental approach. In a commercial/application setting, one would naturally select a different nanolithography approach if cost-effectiveness would be important.
- The synthesis of colloidal nanoparticles may be relatively cheap but it has the downside still that it is hard to scale since there exist large batch-to-batch variations, in particular if modern shape-selected structures are targeted. Furthermore, the yield of the structures with the targeted shape or size within a batch is also often still quite limited. Therefore, a current trend in the colloidal synthesis field is the application of automated flow-synthesis protocols to mitigate both the batch-to-batch variation problem and obtain higher yield of the particles with the desired structural properties.

Minor comments

"electron-beam lithography onto an oxidized silicon wafer (Fig. 1a)". Figure 1a, however, shows transmission electron images: how thick was the silicon wafer? Or are these deposited on a TEM substrate? If so, it should be mentioned. Also, do you expect a different grain structure when depositing on different substrates?

Our reply: The sample, the results for which are shown in Fig. 1a was indeed made on TEM membrane. The other two samples were fabricated on oxidized silicon. We have clarified this point by making the following changes in the main text on page 4:

"electron-beam lithography onto electron transparent silicon nitride membranes or an oxidized silicon wafer (Fig. 1a)".

As well as stated this clearly in the Methods section (Sample fabrication for optical and TEM measurements):

"The sample with 24 particles for optical measurements and samples for TEM characterization were fabricated on square, $150 \times 150 \mu\text{m}$, 40-nm-thick Si_3N_4 membranes supported by bulk silicon on all four sides⁴⁹. The two samples with 180 particles each were prepared on an n-doped (100)-Si wafer of 500 μm thickness (SiMat) with 118 nm dry oxide layer grown at 1050°C in a Centrotherm furnace. The subsequent processing steps were the same for both types of substrates."

We don't expect different grain structure when depositing on these two different substrates, because both SiN_x and thermally grown SiO_2 surfaces are amorphous.

Typo: "and ensembles of Pd nanodisks38 and"

Our reply: Corrected.

The determination of t_{50} crucially depends on the speed at which the pressure changes in the sample chamber. While a reference to the previous publication explaining the setup is given ("To resolve the fast hydrogen sorption kinetics for all single particles simultaneously, we have employed multiplexed single-particle plasmonic nanoimaging microscopy³⁴"), it would be desirable to mention this aspect also here.

Our reply: We agree with the Referee that we have not been clear enough on this point. Specifically, ref. 34 that we cite only concerns the overall principle of *multiplexed* single-particle plasmonic nanoimaging microscopy, it does not in any way discuss the setup used here to achieve the hydrogenation kinetics measurements. We have therefore added the following sentence to the main text on p. 5 to address this point:

"The determination of rate constants in kinetics measurements can depend on the speed at which the pressure changes in the sample chamber. In our experimental setup (SI Fig. 2), we use series of valves at inlet and outlet sides of the measurement chamber, which allow rapid exchange of pressures during both absorption and desorption step with a time constant of < 0.5 s, so that the effect of the speed of the pressure changes on the kinetics is nearly negligible."

A discussion of the meaning and significance of the "sum of square errors (SSE), R-square, adjusted R-square and Root Mean Squared Error (RMSE)" in SI Fig. 10 would be nice. What do these values mean?

Our reply: We have added the following text to SI to clarify these points:

“SSE measures the total deviation of the response values from the fit to the response values. A value closer to 0 indicates that the model has a smaller random error component, and that the fit will be more useful for prediction.

R-square measures how successful the fit is in explaining the variation of the data, or in other words, it is the square of the correlation between the response values and the predicted response values. R-square can take on any value between 0 and 1, with a value closer to 1 indicating that a greater proportion of variance is accounted for by the model.

The adjusted R-square uses the R-square statistic defined above, and adjusts it based on the residual degrees of freedom. The residual degrees of freedom is defined as the number of response values rv minus the number of fitted coefficients fc estimated from the response values ($v = rv - fc$). v indicates the number of independent pieces of information involving the rv data points that are required to calculate the sum of squares. The adjusted R-square statistic can take on any value less than or equal to 1, with a value closer to 1 indicating a better fit.

RMSE is also known as the fit standard error and the standard error of the regression. It is an estimate of the standard deviation of the random component in the data, and just as with SSE, a mean square error value closer to 0 indicates a fit that is more useful for prediction.”

"Deriving the average crystallite size after each step reveals": an explanation of how GIXRD patterns are analysed to extract the crystallite size should be given, or if an explanation would be too trivial, at the very least a reference.

Our reply: We have modified the corresponding text on p.9 as follows:

“Deriving the average crystallite size after each step reveals (see Methods section for details on size extraction procedure) ... ”

In the Methods section (GIXRD measurements) we then have the following description for the extraction of grain size, which is a standard approach:

“The acquired 2D images were reduced using standard protocols with the SAXSGUI software. The FWHM of the Pd 111 reflection at $40^\circ 2\theta$ was determined using a combination of Gaussian and Lorentzian functions, and the instrumental contribution was subtracted to obtain the peak widening from crystallite size effects. The Scherrer equation was then used to estimate the crystallite sizes.”

Why also fitting the cycling evolution of the crystallite size with a logarithmic function, if "the conventional power-law representation for the average radius of a crystallite during grain growth reads as" equation (2)? What do we learn from the logarithmic fit?

Our reply: We thank the Referee for pointing this out. In fact, the logarithmic fit was a “residue” from earlier version of our analysis, which then stayed in the figure for no real reason. Therefore, we have prepared a new version of Figure 5b with only the power-law fit.

"To further corroborate the GIXRD results, we performed transmission electron microscopy (TEM) characterization of a single nanoparticle before cycling, and after 12 and 30 (de)hydrogenation cycles, respectively, which confirms the significant hydrogen sorption-induced grain growth (Fig. 3c).": is this the same particle? Also, why showing both 12 and 30 cycles? They don't seem to really differ in the grain size, as also expected from the trend in Figure 3b.

Our reply: Indeed, the TEM images show the same particle, and since it is the same particle at the different stages of hydrogen cycling treatment, we choose to show all three images. We modified the Figure 3c caption as follows to make it even more clear that it's the same particle:

"(c) TEM image series of the same representative single Pd particle after 0, 12 and 30 (de)hydrogenation cycles, ..."

The histograms in Figure 4a and 4b are very difficult to read. For example, it took me a while to realize that absorption cycles 2 and 3 are overlapping. I suggest plotting the histogram separately in the SI and replace them here with the average and standard deviation for each cycle.

Our reply: We thank the Referee for pointing this out and made changes to the color scheme used in Fig. 4 a, b ((see also pasted below) to now make overlapping histograms more apparent. We want to keep the histograms in Fig. 4 as their purpose is to show *both* the increase in the t_{50} overall *and* in the spread of these values over time. In addition, with the histograms, we would like to emphasize that these results are obtained from each of the individual nanoparticles that we measure and are *not* a simple average of a sample.

Typo: "by taking found the grain-growth"

Our reply: Corrected.

Report of Referee 2

This communication reports on a systematic study of the hydriding kinetics of Pd nanoparticles, with plasmonic microscopy as the main tool. This approach enables tracking the hydriding kinetics of single particles while correlating it with the evolution of their microstructures. The results are truly impressive, in that they provide a quantitative picture of the evolution of both crystallite configuration and hydriding kinetics upon multiple H₂ cycling, and establishes a convincing link between them. Therefore, I think that this paper is suitable for publication in Nature Communications. However, I have some comments which should be addressed, notably concerning the concentration of hydrogen at defect sites under vacuum and the decoupling of grain boundary effects from these of other defects.

Our reply: We thank the Referee recommending our work for publication and the other kind words. We are addressing the constructive specific comments below in detail.

Important general comments:

1) (i) The rapidly slowing kinetics observed by the authors during H₂ absorption and desorption is not something that is systematically observed on similar Pd-based nanomaterials, to the best of my knowledge (I am thinking about thin Pd films in particular).

(ii) While wondering why this could be the case here, I realized that one important experimental aspect of the kinetic study is not discussed at all in the paper: H₂ is simply vacuumed and not exposed to air between the absorption/desorption cycles. Re-exposing the particles to H₂ after vacuuming or venting are two very different situations. Indeed, if one looks for example at Phys. Chem. Chem. Phys. 2011 (13) 11412 (Fig. 2), it was shown for thin Pd films that residual hydrogen remains in the films after vacuuming, notably because part of the H atoms remain trapped in crystalline defects, which are deep potential wells (but venting will always result in a full hydrogen desorption). Obviously, if this is the case, the kinetics of the first absorption will be very different from the second one and could explain what appears as a rapidly slowing kinetics from cycle to cycle. Furthermore, if grain boundaries are evolving between each cycle, so will the amount of trapped hydrogen under vacuum. I don't contest the link that the authors establish between the evolution of crystallite sizes and the kinetics of H₂ absorption/desorption, but I think my argument should be addressed by the authors and an assumption on the concentration of hydrogen after vacuuming in each absorption/desorption cycle should at least be made, e.g. in the experimental part.

Our reply:

(i) We agree with the Referee that similarly rapidly slowing kinetics have not been observed for thin films. As the main important difference between thin film systems and our nanoparticle system at hand, we can notice that the external-surface to bulk volume ratio is much larger in our case, and accordingly the spatial constraints for grain relaxation during their growth are less restrictive. Thus, the grain growth is expected to be much faster in our case.

(ii) While we agree with the Referee that hydrogen is likely to be trapped at defects (i.e., mainly at grain boundaries in our case), the trapped amount is rather low even for the first cycles. Therefore, we do not think that the effect mentioned by the reviewer is very important. Furthermore, we do not really understand why it would lead to a faster first cycle since filling all the defects means that, upon first hydrogen exposure, more hydrogen needs to enter the system and thus be dissociated, which rather should lead to slightly slower kinetics for the first sorption (rather than faster as we observe).

2) (i) *The increase of Pd crystallite size with the number of absorption/desorption cycles is also not something that is commonly observed, at least with bigger crystallites (e.g. from several tens of nanometers like in thin films). I guess this is only happening because the defect density is very high here.*

(ii) *Do the authors expect defect recovery to reach a plateau? From the fit in Fig. 3b for example, do the authors identify a value at which the crystallite size saturates? If so, this could be a valuable information to add to the paper.*

Our reply:

(i) Here we refer to our response to the comment above. In short, yes, it is related to the much higher grain boundary density in our particles due to the very small grain size and also to the fact that they are less constrained than thin films.

(ii) According to the power-law fit that we present in the main text (Eq. 2) with parameter $m = 5$, the grain size will keep slowly growing with increasing number of cycles (see figure below), however in reality it might saturate, at least apparently, i.e., on the timescale of measurement.

3) *The effect of crystallite size on the kinetics of H₂ absorption/desorption seems clear here, but I am not comfortable with the absence of related discussion on the effect of other crystalline defects. I am sure the authors are aware that other crystalline defects (dislocations, vacancies, twins, impurities, etc) also affect the hydriding mechanism (see e.g. A. Pundt and R. Kirchheim, Annu Rev Mater Res 36 (2006) 555). In the best-case scenario, the other defects are not evolving with H₂ cycling, but this is most likely not the case (e.g. in Int. J. Hydrogen Energy 40 (2015) 7335, thin films absorbing hydrogen, even in the alpha-phase, show a significant increase in dislocation density). This is a very complex situation: the density of some defects might increase, decrease or stay constant. First of all, the authors should make assumptions on the behavior of other defects with H₂ cycling, and second, they should explain why they think that the effect of grain boundaries is the main effect studied here. And of course, any experimental evidence to support these assumptions (e.g., defect statistics from ex-situ characterization) are highly valuable.*

Our reply: We agree with the referee, of course, that other defects than grain boundaries are present in Pd systems and that they all interact with hydrogen in their specific ways, as also nicely discussed in the invoked articles. However, as the key difference to the system we investigate, we find the initial grain size, which is in the sub-10 nm range in our case. In other

words, it is questionable that we will find a significant amount of, for example, dislocations in such small grains, since they energetically likely are not allowed for the grain sizes at hand (Matthews, J. W. & Blakeslee, A. E. Journal of Crystal Growth 27, 118-125, (1974), discussion in SI in Syrenova, S. et al. Nature Materials 14, 1236-1244, (2015) and Wagner, S. *et al.* Scripta Materialia 64, 978-981, (2011).). Likewise, the number of vacancies is expected to be low compared to grain-boundary-related defects, again due to the very small grain size in our system. Finally, also impurities are expected to be at a low level since we use a high purity Pd source for the particle fabrication and, most importantly, impurity concentration should stay constant during cycling and therefore affect all cycles reasonably equally. In summary, we thus are convinced that grain-boundary-related defects are the key factor for the observed slowing effect of the kinetics. To make this clear, we have added the following text to the revised manuscript:

“As a final aspect, we also notice that other types of structural defects, such as impurities, vacancies and dislocations have been reported to affect hydrogen sorption kinetics in Pd systems like thin films^{54,55}. As the main difference we find, however, that the initial grain size in our case is significantly smaller, which has a number of key consequences. The first one is that dislocations not associated with grain boundaries, i.e., within the crystallites, are very unlikely to play a role since they energetically are not allowed in crystallites in the (sub-) 10 nm range^{8,56,57}. Similarly, the number of vacancies is expected to be low compared to grain-boundary-related defects and impurities are very unlikely to play a critical role due to the high purity source material used to grow the Pd nanoparticles.”

Specific comments:

1) Page 2, line 5: I dont like the term completely uncharted here. On page 3 line 15, the authors use the term widely lacking which seems more appropriate to me with respect to the state of the art that the authors correctly describe in the introduction.

Our reply: We have reworded to: “poorly understood”

2) Page 4, line 22: The crystalline orientation of the oxidized silicon wafers should be specified.

We have specified oxidized silicon wafer orientation in the Methods section (Sample fabrication for optical and TEM measurements):

“The sample with 24 particles for optical measurements and samples for TEM characterization were fabricated on square, $150 \times 150 \mu\text{m}$, 40-nm-thick Si_3N_4 membranes supported by bulk silicon on all four sides⁴⁹. The two samples with 180 particles each were prepared on an n-doped (100)-Si wafer of 500 μm thickness (SiMat) with 118 nm dry amorphous oxide layer grown at 1050°C in a Centrotherm furnace. The subsequent processing steps were the same for both types of substrates.”

REVIEWER COMMENTS

Reviewer #1 (Remarks to the Author):

The authors have addressed all my comments and replied to all my doubts. I am happy to recommend the current version for publication.

Reviewer #2 (Remarks to the Author):

I am satisfied with the answers formulated by the authors, as well as the modifications brought to the manuscript. I mainly wanted to challenge the authors on the limits of their experimental approach, and I think the latter is valid in light of their arguments. I still think the amount of H₂ trapped in vacuum between the hydrogen absorption/desorption cycles - especially with such an evolving microstructure - might affect the kinetic interpretation, but this is an intrinsic experimental limitation.

So, I think the paper in its present form is acceptable for publication in *Nature Communications*.